# Accurate and Scalable Graph Neural Networks via Message Invariance

**Zhihao Shi** [1], **Jie Wang**[*1], **Zhiwei Zhuang** [1], **Xize Liang** [1], **Bin Li** [1], **Feng Wu** [1],

[1] MoE Key Laboratory of Brain-inspired Intelligent Perception and Cognition,
University of Science and Technology of China

## Abstract

Message passing-based graph neural networks (GNNs) have achieved great success in many real-world applications. For a sampled mini-batch of target nodes, the message passing process is divided into two parts: **m**essage **p**assing between nodes with**in** the **b**atch (MP$_{IB}$) and **m**essage **p**assing from nodes **o**utside the **b**atch to those within it (MP$_{OB}$). However, MP$_{OB}$ recursively relies on higher-order out-of-batch neighbors, leading to an exponentially growing computational cost with respect to the number of layers. Due to the *neighbor explosion*, the whole message passing stores most nodes and edges on the GPU such that many GNNs are infeasible to large-scale graphs. To address this challenge, we propose an accurate and fast mini-batch approach for large graph transductive learning, namely **t**op**o**logical com**p**ensation (TOP), which obtains the outputs of the whole message passing solely through MP$_{IB}$, without the costly MP$_{OB}$. The major pillar of TOP is a novel concept of *message invariance*, which defines *message-invariant transformations* to convert costly MP$_{OB}$ into fast MP$_{IB}$. This ensures that the modified MP$_{IB}$ has the same output as the whole message passing. Experiments demonstrate that TOP is significantly faster than existing mini-batch methods by order of magnitude on vast graphs (millions of nodes and billions of edges) with limited accuracy degradation.

## 1 Introduction

Message passing-based graph neural networks (GNNs) have been successfully applied to many practical applications involving graph-structured data, such as social network prediction (Hamilton et al., 2017; Kipf & Welling, 2017; Deng et al., 2019), chip design Geng et al. (2025b); Bai et al. (2025); Wang et al. (2024b;c), combinatorial optimization Geng et al. (2023a; 2025a); Liu et al. (2025; 2023; 2024); Wang et al. (2023; 2024a); Ling et al. (2024), drug reaction (Do et al., 2019; Duvenaud et al., 2015; Geng et al., 2023b), and recommendation systems (Ying et al., 2018; Fan et al., 2019). The key idea of GNNs is to iteratively update the embeddings of each node based on its local neighborhood. Thus, as these iterations progress, each node embedding encodes more and more information from further reaches of the graph (Hamilton, 2020, Chap. 5).

However, training GNNs on a large-scale graph is challenging due to the well-known *neighbor explosion* problem. Specifically, the embedding of a node at the $l$-th GNN layer depends on the embeddings of its local neighborhood at the $(l-1)$-th GNN layer. Thus, around the target mini-batch nodes, these message passing iterations of an $L$-layer GNN form a tree structure by unfolding their $L$-hop neighborhoods (Hamilton, 2020, Chap. 5), whose size exponentially increases with the GNN depth $L$ (see Figure 1(a)). The exploded source neighborhoods may contain most nodes in the large-scale graph, leading to expensive computational costs (Shi et al., 2024).

To alleviate this problem, recent graph sampling techniques approximate the whole message passing with the small size of the source neighborhoods (Ma & Tang, 2021, Chap. 7). For example, node-wise (Hamilton et al., 2017; Chen et al., 2018a; Balin & Catalyurek, 2023) and layer-wise (Chen et al., 2018b; Zou et al., 2019; Huang et al., 2018) sampling recursively sample a small set of local neighbors over message passing layers. The expectation of the recursive sampling obtains the whole message passing and thus the recursive sampling is accurate and provably convergent (Chen et al., 2018a).

---

*Corresponding author: jiewangx@ustc.edu.cn

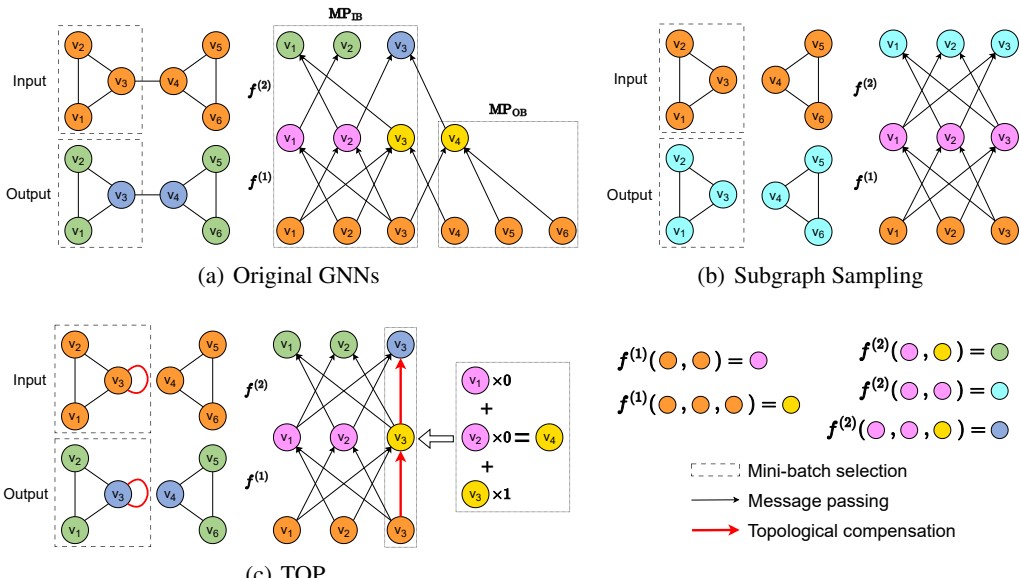

Figure 1: **Mini-batch processing of original GNNs, subgraph sampling, and TOP.** Given a mini-batch, the computational costs of original GNNs exponentially increase with GNN depth (a). To address this challenge, many subgraph sampling methods preserve message passing between the in-batch nodes (MP$_{\text{IB}}$) and eliminate message passing from out-of-batch neighbors to the in-batch nodes (MP$_{\text{OB}}$) to reduce the computational costs (b). However, the final embeddings of subgraph sampling are usually different from the result of the original GNNs. By noticing the message invariance $\mathbf{h}_4 = 0 \cdot \mathbf{h}_1 + 0 \cdot \mathbf{h}_2 + 1 \cdot \mathbf{h}_3$, TOP converts MP$_{\text{OB}}$ $v_4 \to v_3$ into MP$_{\text{IB}}$ $v_3 \to v_3$ without approximation errors in the example (c).

Different from the recursive fashion, subgraph sampling (Chiang et al., 2019; Zeng et al., 2020; Fey et al., 2021; Zeng et al., 2021) adopts a cheap and simple one-shot sampling fashion, i.e., sampling the same subgraph induced by a mini-batch for different GNN layers. It preserves message passing between in-batch nodes (MP$_{\text{IB}}$) and eliminates message passing from out-of-batch neighbors to in-batch nodes (MP$_{\text{OB}}$), achieving a linear complexity with respect to the number of GNN layers.

Nonetheless, accuracy and efficiency are two important but conflicting factors for existing graph sampling techniques. Specifically, accurate recursive sampling maintains the whole message passing at the expense of efficiency, while fast one-shot sampling eliminates MP$_{\text{OB}}$ at the expense of accuracy. This motivates us to develop an accurate and fast mini-batch method for GNNs to approximate the outputs of the whole message passing solely through MP$_{\text{IB}}$ with marginal errors.

In this paper, we first propose a novel concept of *message invariance*, which defines message-invariant transformations to convert MP$_{\text{OB}}$ into MP$_{\text{IB}}$, ensuring that the modified MP$_{\text{IB}}$ has the same output as the whole message passing. Figure 1 shows a motivating example for message invariance, where converting MP$_{\text{OB}}$ $v_4 \to v_3$ to MP$_{\text{IB}}$ $v_3 \to v_3$ (the red edge) does not affect the output of GNNs. Although the resulting subgraphs are different from the original graph, the in-batch embeddings and corresponding computation graphs are always the same. We conduct extensive experiments to show the approximation of message invariance is effective in various real-world datasets (see Section 5.2)

Inspired by the message-invariant transformations, we propose a fast subgraph sampling method, namely **top**ological com**p**ensation (TOP), which is applicable to various real-world graphs. Specifically, TOP estimates the message invariance using a linear transformation, which learns the linear independence between embeddings of the in-batch nodes and their out-of-batch neighbors. In Figure 1, the out-of-batch embedding of $v_4$ is a linear combination of the in-batch embeddings of $(v_1, v_2, v_3)$ with coefficients $(0, 0, 1)$. We estimate the coefficients using a simple and efficient linear regression on sampled basic embeddings (e.g. the embeddings in GNNs with random initialization). We further show that TOP achieves the convergence rate of $\mathcal{O}(\varepsilon^{-4})$ to reach an $\varepsilon$-approximate stationary point (see Theorem 5.1), which is significantly faster than $\mathcal{O}(\varepsilon^{-6})$ of existing subgraph sampling methods (Shi et al., 2023). We conduct extensive experiments on graphs with various sizes to demonstrate that TOP is significantly faster than existing mini-batch methods with limited accuracy degradation (see Figures 3 and 5).

## 2 RELATED WORK

In this section, we discuss some works related to our proposed method.

**Node-wise sampling.** Node-wise sampling (Hamilton et al., 2017; Chen et al., 2018a; Yu et al., 2022) aggregates messages from a subset of uniformly sampled neighborhoods at each GNN layer, which decreases the bases in the exponentially increasing dependencies. The idea is originally proposed in GraphSAGE (Hamilton et al., 2017). VR-GCN (Chen et al., 2018a) further alleviates the bias and variance by historical embeddings, and then shows that its convergence rate to reach an $\varepsilon$-approximate stationary point is $N = \mathcal{O}(\varepsilon^{-4})$, where $N$ denotes the number of iterations in Theorem 2 in (Chen et al., 2018a). GraphFM-IB further alleviates the staleness of the historical embeddings based on the idea of feature momentum. Although the node-wise sampling methods achieve the convergence rate of $\mathcal{O}(\varepsilon^{-4})$, their computational complexity at each step is still exponentially increasing due to the neighborhood explosion issue.

**Layer-wise sampling.** To avoid the exponentially growing computation of node-wise sampling, layer-wise sampling (Chen et al., 2018b; Zou et al., 2019; Huang et al., 2018) samples a fixed number of nodes for each GNN layer and then uses importance sampling (IS) to reduce variance. However, the optimal distribution of IS depends on the up-to-date embeddings, which are expensive. To tackle this problem, FastGCN (Chen et al., 2018b) proposes to approximate the optimal distribution of IS by the normalized adjacency matrix. Adapt (Huang et al., 2018) proposes a learnable sampled distribution to further alleviate the variance. Nevertheless, as the above-mentioned methods sample nodes independently in each GNN layer, the sampled nodes from two consecutive layers may be connected (Zou et al., 2019). Thus, LADIES (Zou et al., 2019) consider the dependency of sampled nodes between layers by one step forward. By combining the advantages of node-wise and layer-wise sampling approaches using Poisson sampling, LABOR (Balin & Catalyurek, 2023) significantly accelerates convergence under the same node sampling budget constraints.

**Subgraph sampling.** Subgraph sampling methods sample a mini-batch and then construct the subgraph based on the mini-batch (Ma & Tang, 2021, Chap. 7). Thus, we can directly run GNNs on the subgraphs. One of the major challenges is to efficiently encode neighborhood information of the subgraph. To tackle this problem, one line of subgraph sampling is to design subgraph samplers to alleviate the inter-connectivity between subgraphs. For example, CLUSTER-GCN (Chiang et al., 2019) propose subgraph samplers based on graph clustering methods (e.g., METIS (Karypis & Kumar, 1998) and Graclus (Dhillon et al., 2007)) and GRAPHSAINT propose edge, node, or random-walk based samplers. SHADOW (Zeng et al., 2021) proposes to extract the $L$-hop neighbors of a mini-batch and then select an important subset from the $L$-hop neighbors. IBMB (Gasteiger et al., 2022) proposes a novel subgraph sampler where the subgraphs are induced by the mini-batches with high influence scores, such as personalized PageRank scores. Another line of subgraph sampling is to design efficient compensation for the messages from the neighborhood based on existing subgraph samplers. For example, GAS (Fey et al., 2021) proposes historical embeddings to compensate for messages in forward passes and LMC (Shi et al., 2023) further proposes historical gradients to compensate for messages in backward passes. GraphFM-OB (Yu et al., 2022) alleviates the staleness of the historical embeddings based on the idea of feature momentum. Besides the traditional optimization algorithm, SubMix (Abu-El-Haija et al., 2023) proposes a novel learning-to-optimize method for subgraph sampling, which parameterizes subgraph sampling as a convex combination of several heuristics and then learns to accelerate the training of subgraph sampling.

## 3 PRELIMINARIES

We first introduce notations in Section 3.1. Then, we introduce graph neural networks and the neighbor explosion issue in Section 3.2.

### 3.1 NOTATIONS

A graph $\mathcal{G} = (\mathcal{V}, \mathcal{E})$ is defined by a set of nodes $\mathcal{V} = \{1, 2, \ldots, n\}$ and a set of edges $\mathcal{E}$ among these nodes. Let $(i, j) \in \mathcal{E}$ denote an edge going from node $i \in \mathcal{V}$ to node $j \in \mathcal{V}$. Let $(\mathcal{B}_1 \rightarrow \mathcal{B}_2)$ denote the set of edges $\{(i, j) | i \in \mathcal{B}_1, j \in \mathcal{B}_2, (i, j) \in \mathcal{E}\}$ from $\mathcal{B}_1$ to $\mathcal{B}_2$. Let $\mathcal{N}_i = \{j \in \mathcal{V} | (i, j) \in \mathcal{E}\}$ denote the neighborhood of node $i$. Let $\mathcal{N}_{\mathcal{B}} = (\cup_{i \in \mathcal{B}} \mathcal{N}_i) \cup \mathcal{B}$ denote the neighborhoods of a mini-batch $\mathcal{B}$ with

itself. Let $\mathcal{N}_{\mathcal{B}}^c = \mathcal{N}_{\mathcal{B}} - \mathcal{B}$ denote the out-of-batch neighbors of the mini-batch $\mathcal{B}$. We recursively define the set of $k$-hop neighborhoods as $\mathcal{N}_{\mathcal{B}}^k = \mathcal{N}_{\mathcal{N}_{\mathcal{B}}^{k-1}}$ with $\mathcal{N}_{\mathcal{B}}^1 = \mathcal{N}_{\mathcal{B}}$. The adjacency matrix is $\mathbf{A} \in \mathbb{R}^{n \times n}$ with $\mathbf{A}_{ij} = 1$ if $(j, i)$ and $\mathbf{A}_{ij} = 0$ otherwise. Given sets $\mathcal{S}_1 = (i_p)_{p=1}^{|\mathcal{S}_1|}, \mathcal{S}_2 = (j_q)_{q=1}^{|\mathcal{S}_2|}$, the submatrix $\mathbf{A}_{\mathcal{S}_1, \mathcal{S}_2}$ satisfies $[\mathbf{A}_{\mathcal{S}_1, \mathcal{S}_2}]_{p,q} = \mathbf{A}_{i_p, j_q}$. For a positive integer $L$, $[\![L]\!]$ denotes $\{1, \ldots, L\}$.

Let the boldface character $\mathbf{x}_i \in \mathbb{R}^{d_x}$ denote the feature of node $i$ with dimension $d_x$. Let $\mathbf{h}_i \in \mathbb{R}^d$ be the $d$-dimensional embedding of the node $i$. Let $\mathbf{X} = (\mathbf{x}_1, \mathbf{x}_2, \ldots, \mathbf{x}_n)^\top \in \mathbb{R}^{n \times d_x}$ and $\mathbf{H} = (\mathbf{h}_1, \mathbf{h}_2, \ldots, \mathbf{h}_n)^\top \in \mathbb{R}^{n \times d}$. We also denote the node features and embeddings of a mini-batch $\mathcal{B} = (i_k)_{k=1}^{|\mathcal{B}|}$ by $\mathbf{X}_{\mathcal{B}} = (\mathbf{x}_{i_1}, \mathbf{x}_{i_2}, \ldots, \mathbf{x}_{i_{|\mathcal{B}|}})^\top \in \mathbb{R}^{|\mathcal{B}| \times d_x}$ and $\mathbf{H}_{\mathcal{B}} \in \mathbb{R}^{|\mathcal{B}| \times d}$ respectively.

## 3.2 GRAPH CONVOLUTIONAL NETWORKS

For simplicity of the derivation, we present our algorithm with graph convolutional networks (GCNs) (Kipf & Welling, 2017). However, our algorithm is also applicable to arbitrary message passing-based GNNs (see Appendix B.1).

A graph convolution layer is defined as

$$\mathbf{H}^{(l+1)} = f^{(l+1)}(\mathbf{H}^{(l)}, \widetilde{\mathbf{A}}) = \sigma(\mathbf{Z}^{(l+1)}\mathbf{W}^{(l)}) = \sigma(\widetilde{\mathbf{A}}\mathbf{H}^{(l)}\mathbf{W}^{(l)}), \ (l+1) \in [\![L]\!], \tag{1}$$

where $\widetilde{\mathbf{A}} = (\mathbf{D}+\mathbf{I})^{-1/2}(\mathbf{A}+\mathbf{I})(\mathbf{D}+\mathbf{I})^{-1/2}$ is the normalized adjacency matrix and $\mathbf{D}$ is the in-degree matrix ($\mathbf{D}_{uu} = \sum_v \mathbf{A}_{uv}$). The initial node feature is $\mathbf{H}^{(0)} = \mathbf{X}$, $\sigma$ is an activation function, and $\mathbf{W}^{(l)}$ is a trainable weight matrix. For simplicity, we denote the GNN parameters $\{\mathbf{W}^{(l)}\}_{l=0}^{L-1}$ by $\mathcal{W}$. Thus, GCNs take node features and the normalized adjacency matrix $(\mathbf{X}, \widetilde{\mathbf{A}})$ as input

$$\mathbf{H}^{(L)} = \text{GCN}(\mathbf{X}, \widetilde{\mathbf{A}}),$$

where $\text{GCN} = f^{(L)} \circ f^{(L-1)} \cdots \circ f^{(1)}$.

The neighbor explosion issue is mainly due to feature propagation $\mathbf{Z}^{(l+1)} = \widetilde{\mathbf{A}}\mathbf{H}^{(l)}$. Specifically, the mini-batch embeddings at the $(l+1)$-th layer

$$\mathbf{H}_{\mathcal{B}}^{(l+1)} = \sigma\left(\mathbf{Z}_{\mathcal{B}}^{(l+1)}\mathbf{W}^{(l)}\right) = \sigma\left(\widetilde{\mathbf{A}}_{\mathcal{B}, \mathcal{N}_{\mathcal{B}}}\mathbf{H}_{\mathcal{N}_{\mathcal{B}}}^{(l)}\mathbf{W}^{(l)}\right) \tag{2}$$

recursively depend on $\mathbf{H}_{\mathcal{N}_{\mathcal{B}}}^{(l)}$ at the $l$-th layer. Thus, the dependencies of nodes (i.e., $\mathbf{H}_{\mathcal{B}}^{(L)}$ depends on $\mathbf{H}_{\mathcal{N}_{\mathcal{B}}^L}^{(0)}$ [1]) are exponentially increasing with respect to the number of layers $L$ due to $\mathcal{O}(|\mathcal{N}_{\mathcal{B}}^L|) = \mathcal{O}(|\mathcal{B}|deg_{\max}^L)$ with the maximum degree $deg_{\max}$.

## 4 MESSAGE INVARIANCE

In this section, we elaborate on message invariance in detail. We first present the definition of message invariance in Section 4.1. We then provide a case study for message invariance in Section 4.2.

## 4.1 MESSAGE INVARIANCE

We first separate the mini-batch feature propagation in Equation (2) into two parts, i.e.,

$$\mathbf{Z}_{\mathcal{B}}^{(l+1)} = \underbrace{\widetilde{\mathbf{A}}_{\mathcal{B}, \mathcal{B}}\mathbf{H}_{\mathcal{B}}^{(l)}}_{\text{MP}_{\text{IB}}} + \underbrace{\widetilde{\mathbf{A}}_{\mathcal{B}, \mathcal{N}_{\mathcal{B}}^c}\mathbf{H}_{\mathcal{N}_{\mathcal{B}}^c}^{(l)}}_{\text{MP}_{\text{OB}}}, \tag{3}$$

where $\text{MP}_{\text{IB}}$ and $\text{MP}_{\text{OB}}$ denote message passing between the in-batch nodes and message passing from their out-of-batch neighbors to the in-batch nodes respectively.

To avoid the recursive dependencies induced by $\text{MP}_{\text{OB}}$, we first introduce a novel concept of (global) message invariance, which bridges the gap between costly $\text{MP}_{\text{OB}}$ and fast $\text{MP}_{\text{IB}}$.

**Definition 4.1** (Message invariance). We say that a transformation $g : \mathbb{R}^{|B| \times d} \to \mathbb{R}^{|\mathcal{N}_{\mathcal{B}}^c| \times d}$ is message-invariant if it satisfies

$$\mathbf{H}_{\mathcal{N}_{\mathcal{B}}^c}^{(l)} = g(\mathbf{H}_{\mathcal{B}}^{(l)}). \tag{4}$$

for any GNN parameters $\mathcal{W}$.

---

[1] $\mathcal{N}_{\mathcal{B}}^L = \|[\widetilde{\mathbf{A}}^L]_{\mathcal{B}}\|_0.$

Given the message invariance, the composition of the original $\text{MP}_{\text{OB}}$ operator $\widetilde{\mathbf{A}}_{\mathcal{B},\mathcal{N}_{\mathcal{B}}^c} : \mathbb{R}^{|\mathcal{N}_{\mathcal{B}}^c| \times d} \to \mathbb{R}^{|\mathcal{B}| \times d}$ and the transformation $g : \mathbb{R}^{|\mathcal{B}| \times d} \to \mathbb{R}^{|\mathcal{N}_{\mathcal{B}}^c| \times d}$ leads to a new $\text{MP}_{\text{IB}}$ operator $(\widetilde{\mathbf{A}}_{\mathcal{B},\mathcal{N}_{\mathcal{B}}^c} g) : \mathbb{R}^{|\mathcal{B}| \times d} \to \mathbb{R}^{|\mathcal{B}| \times d}$. Thus, the mini-batch feature propagation (3) becomes

$$\mathbf{Z}_{\mathcal{B}}^{(l+1)} = \underbrace{\widetilde{\mathbf{A}}_{\mathcal{B},\mathcal{B}} \mathbf{H}_{\mathcal{B}}^{(l)}}_{\text{MP}_{\text{IB}}} + \underbrace{\widetilde{\mathbf{A}}_{\mathcal{B},\mathcal{N}_{\mathcal{B}}^c} g(\mathbf{H}_{\mathcal{B}}^{(l)})}_{\text{MP}_{\text{IB}}}, \tag{5}$$

which is independent of the neighborhood embeddings $\mathbf{H}_{\mathcal{N}_{\mathcal{B}}^c}^{(l)}$. Therefore, the message-invariant transformation $g$ avoids the recursive dependencies and expensive costs of out-of-batch neighbors.

## 4.2 A Case Study for Message Invariance

Due to the arbitrariness of graph structures and the nonlinearity of GNNs, the formula of the message-invariant transformation $g$ is usually unknown. Here we provide a case study for a specific form of $g$ by simplifying the graph structures or the GNN architectures. The case study will motivate us to estimate the message-invariant transformation $g$ in Section 5.1.

### 4.2.1 Message Invariance on Graph with Symmetry

The first example is shown in Figure 1, where the node features are finite and the GNN architectures are arbitrary. Due to the permutation equivariance of GNNs, the nodes in the graph are categorized into two sets $S_1 = \{v_1, v_2, v_5, v_6\}$ and $S_2 = \{v_3, v_4\}$, where the nodes in the same set are isomorphic to each other. The embeddings of isomorphic nodes are always the same, regardless of the GNN architectures. Therefore, the message-invariant transformation is

$$\mathbf{h}_4^{(l)} = g(\mathbf{h}_1^{(l)}, \mathbf{h}_2^{(l)}, \mathbf{h}_3^{(l)}) = 0 \cdot \mathbf{h}_1^{(l)} + 0 \cdot \mathbf{h}_2^{(l)} + 1 \cdot \mathbf{h}_3^{(l)}.$$

Notably, the selection of mini-batches does not require considering the symmetry of the graph in Figure 1. If the mini-batch $\mathcal{B}$ consists of two nodes $v_2$ and $v_3$ from $S_1$ and $S_2$ respectively, then finding $g$ is still easy by $\mathbf{h}_1^{(l)} = 1 \cdot \mathbf{h}_2^{(l)} + 0 \cdot \mathbf{h}_3^{(l)}$ and $\mathbf{h}_4^{(l)} = 0 \cdot \mathbf{h}_2^{(l)} + 1 \cdot \mathbf{h}_3^{(l)}$. In the example, the condition for the existence of the message-invariant transformation is that the mini-batch $\mathcal{B}$ contains at least one node from each of $S_1$ and $S_2$.

The example discusses a small graph with six nodes, while many real-world graphs contain millions of nodes. From a probabilistic perspective, the sets $S_1$ and $S_2$ represent two peaks of the data distribution. Then, the condition becomes that the mini-batch $\mathcal{B}$ contains the most frequent node inputs (the node features and their neighborhood structures). These frequent node inputs are also sampled with a high probability under a large enough batch size. Thus, the message-invariant transformation is easy to find in large-scale graphs. We provide the detailed formulation and theory in Appendix E.

### 4.2.2 Message Invariance for Linear GNNs

We use linear GNNs (Xu et al., 2021; Wang & Zhang, 2022) as the second example, which simplifies the GNN architectures without restricting the graph structures. Linear GNNs use an identity mapping $\sigma$ as the activation function. For linear GNNs $\mathbf{H}^{(l)} = \widetilde{\mathbf{A}}^l \mathbf{X} \mathbf{W}^{(0)} \dots \mathbf{W}^{(l-1)}$, the linear dependence between embeddings $\mathbf{H}^{(l)}$ is equal to the linear dependence between the corresponding parameter-free features $\widetilde{\mathbf{A}}^l \mathbf{X}$. Specifically, if the $l$-hop features $\mathbf{X}_{\mathcal{B}}^{(l)} = (\widetilde{\mathbf{A}}^l \mathbf{X})_{\mathcal{B}}$ is a full-column-rank matrix, then there exists a coefficient matrix $\mathbf{R}$ such that $\mathbf{X}_{\mathcal{N}_{\mathcal{B}}^c}^{(l)} = \mathbf{R} \mathbf{X}_{\mathcal{B}}^{(l)}$. Then, the linear dependence between embeddings is

$$\mathbf{H}_{\mathcal{N}_{\mathcal{B}}^c}^{(l)} = \mathbf{X}_{\mathcal{N}_{\mathcal{B}}^c}^{(l)} \mathbf{W}^{(0)} \dots \mathbf{W}^{(l-1)} = \mathbf{R} \mathbf{X}_{\mathcal{B}}^{(l)} \mathbf{W}^{(0)} \dots \mathbf{W}^{(l-1)} = \mathbf{R} \mathbf{H}_{\mathcal{B}}^{(l)}.$$

Thus, the message-invariant transformation $g$ in Equation (4) is a linear transformation for the coefficient matrix $\mathbf{R}$.

For non-linear GNNs, the relation between embeddings of the in-batch nodes and their out-of-batch neighbors may be non-linear. Nonetheless, on the real-world datasets, the linear message-invariant transformation has achieved marginal approximation errors in practice as shown in Section 5.2.

# 5 TOPOLOGICAL COMPENSATION

In this section, we present the details of the proposed topological compensation framework (TOP). First, we introduce the formulation of TOP inspired by the case study of message invariance in Section 5.1. Then, based on the linear estimation of TOP, we conduct experiments to demonstrate that the message invariance significantly reduces the discrepancy between $MP_{IB}$ and the whole message passing in Section 5.2. Finally, we analyze the convergence of TOP in Section 5.3.

## 5.1 FORMULATION OF TOPOLOGICAL COMPENSATION

**Formulation.** Inspired by the linear message-invariant transformation in the case study in Section 4.2, we propose to model message invariance $\mathbf{H}_{\mathcal{N}_{\mathcal{B}}^c}^{(l)}$ by $\mathbf{H}_{\mathcal{N}_{\mathcal{B}}^c}^{(l)} \approx \mathbf{R}\mathbf{H}_{\mathcal{B}}^{(l)}$, where the coefficient matrix $\mathbf{R} \in \mathbb{R}^{|\mathcal{N}_{\mathcal{B}}^c| \times |\mathcal{B}|}$ are the weights of linear combinations of the in-batch embeddings of $\mathbf{H}_{\mathcal{B}}^{(l)}$. Combining the approximation and the mini-batch feature propagation (3) leads to

$$\mathbf{Z}_{\mathcal{B}}^{(l+1)} \approx \widetilde{\mathbf{A}}_{\mathcal{B},\mathcal{B}}\mathbf{H}_{\mathcal{B}}^{(l)} + \widetilde{\mathbf{A}}_{\mathcal{B},\mathcal{N}_{\mathcal{B}}^c}\mathbf{R}\mathbf{H}_{\mathcal{B}}^{(l)} = \underbrace{(\widetilde{\mathbf{A}}_{\mathcal{B},\mathcal{B}} + \partial\mathbf{A}_{\mathcal{B},\mathcal{B}})\mathbf{H}_{\mathcal{B}}^{(l)}}_{MP_{IB}}, \tag{6}$$

where we call $\partial\mathbf{A}_{\mathcal{B},\mathcal{B}} \triangleq \widetilde{\mathbf{A}}_{\mathcal{B},\mathcal{N}_{\mathcal{B}}^c}\mathbf{R}$ *the topological compensation* (TOP). The topological compensation implements the message invariance by adding weighted edges to the induced subgraph $\widetilde{\mathbf{A}}_{\mathcal{B},\mathcal{B}}$. Then, TOP directly runs a GCN on the modified subgraph as follows

$$\mathbf{H}_{\mathcal{B}}^{(L)} = \text{GCN}(\mathbf{X}_{\mathcal{B}}, \widetilde{\mathbf{A}}_{\mathcal{B},\mathcal{B}} + \partial\mathbf{A}_{\mathcal{B},\mathcal{B}}).$$

The formulation of TOP makes it easy to incorporate the existing subgraph sampling methods.

**Estimation of topological compensation.** To reduce the discrepancy between the modified $MP_{IB}$ in Equation (6) and the whole message passing (3), we estimate $\mathbf{R}$ by

$$\min_{\mathbf{R}} \|\mathbf{R}\overline{\mathbf{H}}_{\mathcal{B}} - \overline{\mathbf{H}}_{\mathcal{N}_{\mathcal{B}}^c}\|_F,$$

where $\overline{\mathbf{H}}$ denotes the basic embeddings and $\|\cdot\|_F$ is the Frobenius norm. The basic embeddings reflect the similarity between nodes.

**Selection of basic embeddings.** Before the training, we select the basic embeddings of a GNN at random initialization by $\overline{\mathbf{H}}(\mathcal{W}^{(rand)}) = (\mathbf{H}^{(0,rand)}, \mathbf{H}^{(1,rand)}, \ldots, \mathbf{H}^{(T,rand)}) \in \mathbb{R}^{n \times (T+1)d}$, where $\mathcal{W}^{(rand)}$ are the randomly initialized parameters and $\mathbf{H}^{(j,rand)}$ are the corresponding embeddings at the $j$-th layer. The basic embeddings are the concatenation of all embeddings at different layers.

An appealing feature of $\overline{\mathbf{H}}(\mathcal{W}^{(rand)})$ is that they can identify the 1-WL indistinguishable node pairs by Theorem E.3 in Appendix E. The property ensures that the learned $g$ is message-invariant on graphs with symmetry or large-scale graphs like the first motivating example in Section 4.2.1.

The linear message-invariant transformation with the basic embeddings $\overline{\mathbf{H}}(\mathcal{W}^{(rand)})$ is very accurate on real-world datasets as shown in Section 5.2. Thus, we estimate TOP in the pre-processing phase and then reuse it during the training phase for efficiency in our experiments. When TOP based on $\overline{\mathbf{H}}(\mathcal{W}^{(rand)})$ suffers from high errors, a solution is to update $g$ using the up-to-date embeddings $\overline{\mathbf{H}}(\mathcal{W}^{(t)})$ at the $t$-th training step.

## 5.2 MEASURING MESSAGE INVARIANCE IN REAL-WORLD DATASETS.

In this section, we conduct experiments to demonstrate that the message invariance significantly reduces the discrepancy between $MP_{IB}$ and the whole message passing in many real-world datasets. To ensure the robustness and generalizability of TOP in practice, we provide more results in Tables 3, 7, and 8, including more experiments on heterophilous graphs and experiments under various subgraph samplers. The whole experiments are conducted on five GNN models (GCN, GAT, SAGE, GCNII, and PNA) and eight datasets (Ogbn-arxiv, Reddit, Yelp, Ogbn-products, amazon-ratings, minesweeper, questions, and questions).

**Measuring message invariance in real-world datasets.** We first train GNNs by the full-batch gradient descent for each dataset. Then, we measure the discrepancy between $MP_{IB}$ and the whole

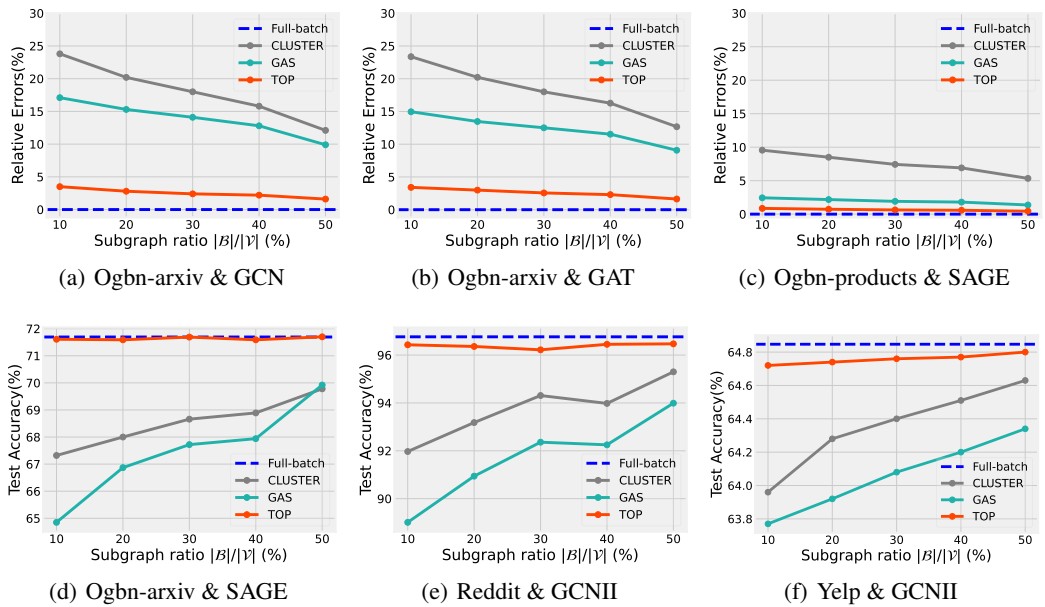

Figure 2: **Measuring the message invariance in real-world datasets.** The output of TOP is very close to the whole message passing (denoted by Full-batch). Please refer to Table 3 in Appendix C.1 for more results.

message passing (denoted by Full-batch) by relative approximation errors and accuracy degradation. The relative approximation errors and accuracy degradation are defined by

$$\frac{\sqrt{\left(\sum_{i=1}^{b}\|\mathbf{H}_{\mathcal{B}_i}^{(L,*)} - \mathbf{H}_{\mathcal{B}_i}^{(L)}\|_F^2\right)}}{\|\mathbf{H}^{(L,*)}\|_F} \quad \text{and} \quad \frac{1}{b}\sum_{i=1}^{b} \mathrm{acc}(\mathbf{H}_{\mathcal{B}_i}^{(L,*)}, \mathbf{Y}_{\mathcal{B}_i}) - \mathrm{acc}(\mathbf{H}_{\mathcal{B}_i}^{(L)}, \mathbf{Y}_{\mathcal{B}_i}),$$

where we run the whole message passing (i.e., Full-batch) to obtain $\mathbf{H}_{\mathcal{B}_i}^{(L,*)}$ and $\mathbf{Y}$ is the matrix consisting of the node labels. We partition the graph into 200 clusters and then sample $b$ in $\{20, 40, 60, 80, 100\}$ clusters to construct subgraphs. If we decrease the batch size $b$, then the ratio of messages in $\mathrm{MP_{OB}}$ increases and thus $\mathrm{MP_{OB}}$ becomes important.

Our baselines include two subgraph sampling methods using $\mathrm{MP_{IB}}$ (i.e., CLUSTER (Chiang et al., 2019) and GAS (Fey et al., 2021)). We introduce these baselines in Appendix A. We report the test accuracy vs. subgraph ratio in Figure 2. The relative approximation errors of TOP are less than 5% and the test accuracy of TOP is very close to Full-batch under different batch sizes.

## 5.3 CONVERGENCE OF TOP

Based on message invariance (4), we develop the convergence analysis of TOP in this section. The assumption of Theorem 5.1 is widely used in convergence analysis (Shi et al., 2023; Chen et al., 2018a; Yu et al., 2022). All proofs are provided in Appendix D.

**Theorem 5.1.** *Let $\mathcal{L}(\mathcal{W}) = \sum_{i \in \mathcal{V}} \ell(\mathbf{h}_i^{(L)}, y_i)/|\mathcal{B}|$ and $\mathbf{d}_{\mathcal{W}} = \nabla_{\mathcal{W}} \sum_{i \in \mathcal{B}} \ell(\mathbf{h}_i^{(L,TOP)}, y_i)/|\mathcal{B}|$ be the loss of the full-batch method and the gradient of TOP respectively, where $\ell$ is the loss function and $y_i$ is the label of node $i$. Assume that (1) the optimal value $\mathcal{L}^* = \inf \mathcal{L}(\mathcal{W})$ is finite (2) at the $k$-th iteration, a batch of nodes $\mathcal{V}_{\mathcal{B}}^k$ is uniformly sampled from $\mathcal{V}$ (3) function $\nabla_{\mathcal{W}} \mathcal{L}$ is $\gamma$-Lipschitz with $\gamma > 1$ (4) norms $\|\nabla_{\mathcal{W}} \mathcal{L}\|_2$ and $\|\mathbf{d}_{\mathcal{W}}\|_2$ are bounded by $G > 1$. With the learning rate $\eta = \mathcal{O}(\varepsilon^2)$ and the training step $N = \mathcal{O}(\varepsilon^{-4})$, TOP then finds an $\varepsilon$-stationary solution such that $\mathbb{E}[\|\nabla_{\mathcal{W}} \mathcal{L}(\mathcal{W}^{(R)})\|_2] \leq \varepsilon$ after running for $N$ iterations, where $R$ is uniformly selected from $[\![N]\!]$.*

The convergence rate $N = \mathcal{O}(\varepsilon^{-4})$ is the same as the standard SGD (Nesterov, 2013; Fang et al., 2018). Notably, the convergence rate of TOP is faster than that of LMC (Shi et al., 2023) (i.e., $N = \mathcal{O}(\varepsilon^{-6})$), as TOP avoids the staleness issue of the historical embeddings and gradients of LMC.

Table 1: **Statistics of the datasets in our experiments**. "#" denotes the number and "Avg. degree" denotes the average degree. The task is node classification, which is a standard task to evaluate the scalability on the large-scale graph (Chiang et al., 2019; Zeng et al., 2020; Fey et al., 2021).

| Dataset | #Classes | Total #Nodes | Total #Edges | Avg. degree | Train/Val/Test |
|---------|----------|--------------|--------------|-------------|----------------|
| Reddit | 41 | 232,965 | 11,606,919 | 49.8 | 0.660/0.100/0.240 |
| Yelp | 50 | 716,847 | 6,997,410 | 9.8 | 0.750/0.150/0.100 |
| Ogbn-arxiv | 40 | 169,343 | 1,157,799 | 6.9 | 0.537/0.176/0.287 |
| Ogbn-products | 47 | 2,449,029 | 61,859,076 | 25.3 | 0.100/0.020/0.880 |
| Ogbn-papers100M | 172 | 111,059,956 | 1,615,685,872 | 14.6 | 0.780/0.080/0.140 |

## 6 EXPERIMENTS

We first compare the convergence and efficiency of TOP with the state-of-the-art subgraph sampling methods—which are the most related baselines—in Section 6.1. Then, we compare the convergence and efficiency of TOP with the state-of-the-art node/layer-wise sampling methods in Section 6.2. More experiments are provided in Appendix C.

### 6.1 COMPARISON WITH SUBGRAPH SAMPLING

**Datasets.**  We evaluate TOP on five datasets with various sizes (i.e., Reddit (Hamilton et al., 2017), Yelp (Zeng et al., 2020), Ogbn-arxiv, Ogbn-products, and Ogbn-papers (Hu et al., 2020)). These datasets contain at least 100 thousand nodes and one million edges. Notably, Ogbn-papers is very large, containing 100 million nodes and 1.6 billion edges. They have been widely used in previous works (Fey et al., 2021; Zeng et al., 2020; Hamilton et al., 2017; Chiang et al., 2019; Chen et al., 2018a;b). Table 1 summarizes the statistics of the datasets. We also conduct experiments on heterophilous graphs in Appendix C.7.

**Subgraph samplers.**  On the small and medium datasets (i.e., Ogbn-arxiv, Reddit, and Yelp), we follow CLUSTER (Chiang et al., 2019) and GAS (Fey et al., 2021) to sample subgraphs based on METIS (see Appendix A.1). Specifically, we first use METIS to partition the original graph into many clusters and then sample a cluster of nodes to generate a subgraph. On the large datasets (i.e., Ogbn-products and Ogbn-papers), as the METIS algorithm is too time-consuming (Zeng et al., 2020), we uniformly sample nodes to construct subgraphs. More experiments under various subgraph samplers are provided in Appendix C.8.

**Baselines and implementation details.**  Our baselines include subgraph sampling (CLUSTER (Chiang et al., 2019), SAINT (Zeng et al., 2020), and GAS (Fey et al., 2021)). We also compare TOP with IBMB (Gasteiger et al., 2022) in Appendix C.3, a recent subgraph sampling method focused on the design of subgraph samplers, which is orthogonal to TOP (see Section 2). We implement TOP, CLUSTER, SAINT, and GAS based on the codes and toolkits of GAS (Fey et al., 2021) to ensure a fair comparison. We introduce these baselines in Appendix A. We evaluate CLUSTER, GAS, SAINT, and TOP based on the same GNN backbone, including the widely used GCN (Kipf & Welling, 2017) and GCNII (Chen et al., 2020). We implement GCN and GCNII following (Fey et al., 2021) and (Hamilton et al., 2017). Due to space limitation, we present the results with more GNN backbones (e.g. SAGE (Hamilton et al., 2017), and GAT (Veličković et al., 2018)) in Appendix C.3. We run all experiments in this section on a single GeForce RTX 2080 Ti (11 GB), and Intel Xeon CPU E5-2640 v4. For other implementation details, please refer to Appendix B.

Figure 3 shows the convergence curves (test accuracy vs. runtime (s)) of TOP, CLUSTER, GAS, SAINT, and Full-batch (i.e. full-batch gradient descent with the whole message passing). We provide the convergence curves (test accuracy vs. epochs) in Appendix C. We use a sliding window to smooth the curves in Figure 3 as the test accuracy is unstable. We ran each experiment five times. The solid curves correspond to the mean, and the shaded regions correspond to values within plus or minus one standard deviation of the mean. The convergence curves consider the runtime of pre-processing.

**Results on small datasets.**  On the small datasets (i.e., Ogbn-arxiv and Reddit), the subgraph ratio $|\mathcal{B}|/|\mathcal{V}|$ is up to 50%, where $|\mathcal{B}|$ and $|\mathcal{V}|$ denote the sizes of subgraphs and the whole graph respectively. The large ratio shows that the subgraph contains much information about the whole graph. According to Figure 3(a), TOP is significantly faster than Full-batch, CLUSTER, GAS, and SAINT without sacrificing accuracy. Further, TOP stably resembles the full-batch performance on the Ogbn-arxiv and Reddit datasets, while CLUSTER, GAS, and SAINT are unstable. The standard

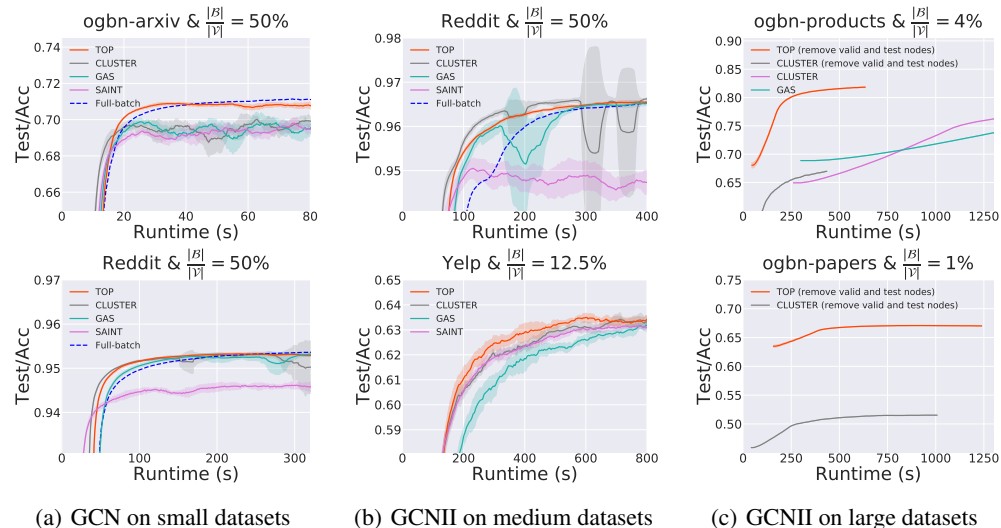

(a) GCN on small datasets    (b) GCNII on medium datasets    (c) GCNII on large datasets

Figure 3: **Convergence curves (test accuracy vs. runtime (s)) of subgraph sampling**. We use the default $|\mathcal{B}|$ and $|\mathcal{V}|$—which denote the sizes of subgraphs and the whole graph respectively—provided in GAS Fey et al. (2021).

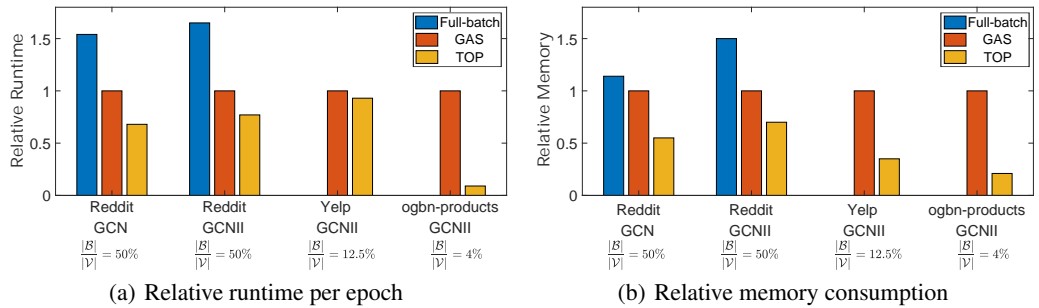

(a) Relative runtime per epoch    (b) Relative memory consumption

Figure 4: **Relative runtime per epoch and relative memory consumption**. Please refer to Table 4 in Appendix C.4 for more results.

deviation of CLUSTER, GAS, and SAINT is large such that the mean test accuracy is lower than the full-batch performance, as they are difficult to encode all available neighborhood information of the subgraph. Specifically, CLUSTER and SAINT do not take MP$_{OB}$ into consideration and GAS uses stale historical embeddings to approximate MP$_{OB}$.

**Results on medium datasets.**   On the medium datasets (i.e., Reddit, and Yelp), the subgraph ratio $|\mathcal{B}|/|\mathcal{V}|$ decreases from 50% to 12.5% due to GPU memory limitations. Thus, Full-batch runs out of GPU memory on the Yelp dataset. Compared with GCN, the nonlinearity of GCNII becomes strong due to the large model capacity of GCNII. Under the strong nonlinearity, TOP is still significantly faster than CLUSTER, GAS, and SAINT on the Yelp dataset with a low subgraph ratio $|\mathcal{B}|/|\mathcal{V}|$ according to Figure 3(b). Moreover, TOP is significantly faster than GAS and Full-batch on the Reddit dataset. Although the mean convergence curses of TOP and CLUSTER are similar on the Reddit dataset, the low standard deviation demonstrates that TOP is more stable than CLUSTER.

**Results on large datasets.**   On the large datasets (i.e., Ogbn-products, and Ogbn-papers), the subgraph ratio $|\mathcal{B}|/|\mathcal{V}|$ is very low due to GPU memory limitations. By noticing that the large number of valid and test nodes in the large datasets is useless for TOP, we remove the valid and test nodes from sampled subgraphs. For an ablation study, we report CLUSTER without valid and test nodes in the sampled subgraphs. We do not remove the valid and test nodes for GAS, as GAS requires updating the historical embeddings on the valid and test nodes to alleviate the staleness issue. Due to a large number of historical embeddings, GAS runs out of CPU memory on the Ogbn-papers dataset. According to Figure 3(c), TOP is significantly faster than CLUSTER and GAS by several orders of magnitude. Moreover, the valid and test nodes in subgraphs are important for CLUSTER, as these valid and test nodes are likely to be the neighbors of the training nodes. The valid and test nodes in

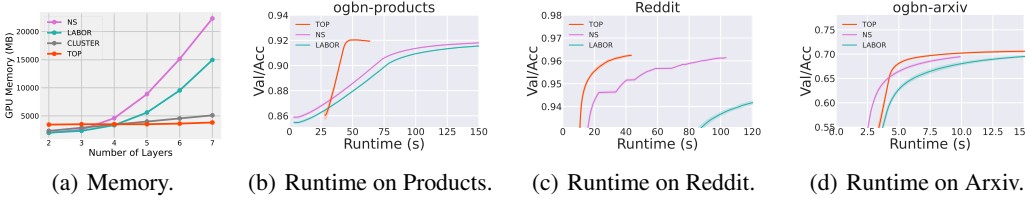

| (a) Memory. | (b) Runtime on Products. | (c) Runtime on Reddit. | (d) Runtime on Arxiv. |

Figure 5: **Memory consumption and convergence curves of TOP and node/layer-wise sampling**.

subgraphs increase the ratio of messages in $\text{MP}_{\text{IB}}$ for CLUSTER. TOP does not depend on the valid and test nodes due to its effective topological compensation.

**Memory and runtime.** We report the GPU memory consumption and the runtime per epoch in Figure 4. TOP is significantly faster and more memory-efficient than GAS on all datasets, as TOP does not require pulling and pushing historical embeddings frequently. Especially, the speedup of TOP against GAS is up to 11x on the Ogbn-product dataset, which is one order of magnitude. We analyze the computational complexity of TOP in Appendix B.6 and give the detailed costs of pre-processing and training in Appendix C.5.

## 6.2 COMPARISON WITH NODE/LAYER-WISE SAMPLING

**Baselines and implementation details** We compare TOP with node-wise and layer-wise sampling methods including neighbor sampling (NS) (Hamilton et al., 2017) and LABOR (Balin & Catalyurek, 2023) in Figure 5, where LABOR combines the advantages of node-wise and layer-wise sampling to accelerate convergence. Unlike subgraph sampling, node/layer-wise sampling mainly focuses on the certain SAGE model (Hamilton et al., 2017). We run NS and LABOR by the official implementation of LABOR (Balin & Catalyurek, 2023). The reported runtime includes the runtime of pre-processing. We run experiments in this section on a single A800 card.

**Hyperparameters.** For TOP, we uniformly sample nodes to construct subgraphs. To ensure a fair comparison, TOP follows the GNN architectures, data splits, training pipeline, learning rate, and hyperparameters of LABOR (Balin & Catalyurek, 2023). We adjust the batch size of TOP such that the memory consumption of TOP is similar to LABOR.

**Memory.** We first evaluate the GPU memory consumption in terms of the number of GNN layers in Figure 5(a). We increase the number of GNN layers from two to seven. The GPU memory of NS and LABOR increases exponentially with the number of GNN layers, and thus they are difficult to apply to deep GNNs (e.g. GCNII with six layers in Figure 3). The GPU memory of both CLUSTER (Chiang et al., 2019) and TOP increases linearly with the number of GNN layers, corresponding with the computational complexity in Table 2. The GPU memory of CLUSTER is slightly larger than TOP, as CLUSTER uses the layer-wise inference (like GAS, see Appendix A.3) in the evaluation phase while TOP only uses the mini-batch information (see Equation (6)).

**Convergence curves.** We further report the convergence curves of TOP, NS, and LABOR in Figures 5(b), 5(c), and 5(d). We have included the pre-processing time of TOP in the figures. Although NS and LABOR do not require pre-processing, TOP finally outperforms NS and LABOR due to its powerful convergence. The speedup of TOP against NS and LABOR is more than 2x on all datasets.

## 7 CONCLUSION

In this paper, we propose an accurate and fast subgraph sampling method, namely topological compensation (TOP), based on a novel concept of message invariance. Message invariance defines message-invariant transformations that convert expensive message passing acted on out-of-batch neighbors ($\text{MP}_{\text{OB}}$) into efficient message passing acted on in-batch nodes ($\text{MP}_{\text{IB}}$). Based on the message invariance, the proposed TOP uses efficient $\text{MP}_{\text{IB}}$ with limited performance degradation. Experiments demonstrate that TOP is significantly faster than existing mini-batch methods by order of magnitude on vast graphs (millions of nodes and billions of edges) with limited performance degradation. While our experiments focus on message-invariant transformation for some common and simple GNNs, non-linear message-invariant transformation needs to be empirically evaluated for more GNNs with more complex aggregation. In the future, we plan to generalize our ideas to more GNNs or graph transformers with global communication.

## REPRODUCIBILITY STATEMENT

To ensure reproducibility, we provide key information from the main text and Appendix as follows.

1. **Algorithm.** We provide the pseudocode of TOP in Algorithms 1 and 2. We also provide the detailed implementation of TOP in Appendix B. See Appendix B.3 for the hyperparameters of TOP.

2. **Theoretical Proofs.** We provide all proofs in Appendix D.

3. **Source Code.** The code of TOP is available on GitHub at `https://github.com/MIRALab-USTC/TOP`.

4. **Experimental Details.** We provide the detailed experimental settings in Section 6 and Appendix B.

## ACKNOWLEDGMENTS

The authors would like to thank all the anonymous reviewers for their valuable suggestions. This work was supported by the National Key R&D Program of China under contract 2022ZD0119801 and the National Nature Science Foundations of China grants U23A20388 and 62021001.

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

## A  BACKGROUND OF SUBGRAPH SAMPLING

Subgraph sampling is a general mini-batch framework for a wide range of GNN architectures. For example, subgraph sampling directly runs a GCN on the subgraph induced by a mini-batch $\mathcal{B}$

$$\mathbf{H}_{\mathcal{B}}^{(L)} \approx \text{GCN}(\mathbf{X}_{\mathcal{B}}, \text{Norm}(\mathbf{A}_{\mathcal{B},\mathcal{B}})),$$

where $\text{Norm}(\cdot)$ normalizes the adjacency matrix of the subgraph $\mathbf{A}_{\mathcal{B},\mathcal{B}}$. For example, CLUSTER-GCN (Chiang et al., 2019) and GraphSAINT (Zeng et al., 2020) use $\text{Norm}(\mathbf{A}_{\mathcal{B},\mathcal{B}}) = \widetilde{\mathbf{A}_{\mathcal{B},\mathcal{B}}}$ and $\text{Norm}(\mathbf{A}_{\mathcal{B},\mathcal{B}}) = \mathbf{D}_{\mathcal{B},\mathcal{B}}^{-1}\mathbf{A}_{\mathcal{B},\mathcal{B}}$ respectively.

Compared with the whole message passing in Equation (1), subgraph sampling drops the edges $\mathcal{N}_{\mathcal{B}}^c \to \mathcal{B}$ from the original graph $\mathbf{A}_{\mathcal{B},\mathcal{N}_{\mathcal{B}}}$, leading to significant approximation errors. Thus, the mini-batch selection of subgraph sampling aims to *minimize the graph cut* from a topological similarity perspective, i.e.,

$$\min_{\mathcal{B}} \|(\mathbf{A}_{\mathcal{B},\mathcal{B}}, \mathbf{O}) - \mathbf{A}_{\mathcal{B},\mathcal{N}_{\mathcal{B}}}\|_0 = \min_{\mathcal{B}} |\mathcal{N}_{\mathcal{B}}^c|, \tag{7}$$

where $\mathbf{O} \in \mathbb{R}^{|\mathcal{B}| \times |\mathcal{N}_{\mathcal{B}}^c|}$ is a zero matrix. Notably, as a connected graph cannot be divided into two disjointed subgraphs without dropping edges, the optimal value of (7) is always positive in the connected graph.

To minimize the graph cut $|\mathcal{N}_{\mathcal{B}}^c|$, the cluster-based samplers (Chiang et al., 2019; Fey et al., 2021; Shi et al., 2023; Yu et al., 2022) first adopt graph clustering (e.g., METIS (Karypis & Kumar, 1998) and Graclus (Dhillon et al., 2007)) to partition the large-scale graph into $\{\mathcal{B}_1, \mathcal{B}_2, \ldots, \mathcal{B}_n\}$ with small $|\mathcal{N}_{\mathcal{B}_i}^c|$ and then sample a subgraph induced by $\mathcal{B}_i$. Besides, the random-walk based sampler (Zeng et al., 2020) first uniformly samples root nodes and then generates random walks $\mathcal{B}$ starting from the root nodes, which decreases the graph cut (Ma & Tang, 2021, Chap. 7).

### A.1  METIS

METIS is a widely used graph clustering technique (Chiang et al., 2019; Fey et al., 2021). Graph clustering aims to construct partitions over the nodes in a graph such that intra-links within clusters occur much more frequently than inter-links between different clusters (Karypis & Kumar, 1998). Intuitively, this results that neighbors of a node are located in the same cluster with high probability. METIS minimizes the graph cut from a topological similarity perspective, i.e. Equation (7), to maintain enough information in the original graph, thus reducing the accesses of inaccurate compensation made by the subgraph sampling method, making the computation faster and more accurate.

However, METIS algorithm is too time-consuming (Zeng et al., 2020) on large datasets (e.g. Ogbn-products and Ogbn-papers). Thus, we uniformly sample nodes to construct subgraphs of large datasets.

### A.2  HISTORICAL EMBEDDINGS

GAS (Fey et al., 2021) further compensates for the messages from the out-of-batch neighbors by historical embeddings, which are defined by

$$\mathbf{Z}_{\mathcal{B}}^{(l+1)} \approx \underbrace{\widetilde{\mathbf{A}}_{\mathcal{B},\mathcal{B}}\mathbf{H}_{\mathcal{B}}^{(l)}}_{\text{MP}_{\text{IB}}} + \underbrace{\widetilde{\mathbf{A}}_{\mathcal{B},\mathcal{N}_{\mathcal{B}}^c}\overline{\mathbf{H}}_{\mathcal{N}_{\mathcal{B}}^c}^{(l)}}_{\text{Bias}}, \tag{8}$$

where $\overline{\mathbf{H}}^{(l)}$ are historical embeedings. GAS pulls historical embeddings from RAM or hard drive storage, making it significantly faster and more memory-efficient than the methods computing real up-to-date embeddings.

However, the historical embeddings suffer from large approximation errors due to the staleness issue (Fey et al., 2021; Yu et al., 2022; Shi et al., 2023). Specifically, GAS updates the historical embeddings in each mini-batch average once per epoch and keeps their values between two consecutive updates of the mini-batch historical embeddings. Thus, if the size of the sampled subgraphs is significantly smaller than the whole graph, the update of historical embeddings is infrequent due to very low node sampling probability, leading to large approximation errors of GAS. Moreover, as the number of

the out-of-batch neighbors is more than that of the nodes in the mini-batch subgraph on large-scale graphs (see Table 6 in (Fey et al., 2021)), pulling a large number of historical embeddings is still expensive.

## A.3   Layer-wise Inference in Evaluation Phase

To ensure the exact inference results on the large graphs, graph sampling usually adapts layer-wise inference, which iteratively updates all node embeddings at each layer without dropping edges. Specifically, the nodes are partitioned into $n$ mini-batches with batch size $|\mathcal{B}|$, denoted as $\mathcal{B}_1, \mathcal{B}_2, ..., \mathcal{B}_n$. At the $l$-th layer, layer-wise inference traverses all mini-baches by

$$\mathbf{H}_{\mathcal{B}_i}^{(l+1)} = \text{GCN}(\mathbf{H}_{\mathcal{N}_{\mathcal{B}_i}}^{(l)}, \widetilde{\mathbf{A}}_{\mathcal{B}_i, \mathcal{N}_{\mathcal{B}_i}}), \text{ for } i \in \{1, 2, ..., n\}.$$

Then, layer-wise inference iteratively updates $\mathbf{H}^{(l+1)}$ on the entire graph based on the previous embeddings $\mathbf{H}^{(l)}$.

For each computation within a batch, the input $\mathbf{H}_{\mathcal{N}_{\mathcal{B}_i}}^{(l)}$ is exact, and the adjacency matrix $\widetilde{\mathbf{A}}_{\mathcal{B}_i, \mathcal{N}_{\mathcal{B}_i}}$ aggregates all the neighbor information. Therefore, $\mathbf{H}_{\mathcal{N}_{\mathcal{B}_i}}^{(l+1)}$ is also exact. Since the model is computed layer-wise, each layer's $\mathbf{H}^{(l+1)}$ is exact. As a result, the final output of the model is exact inference results.

Due to the fact that this layer-wise inference requires the computation of a large amount of data beyond the evolution dataset, it can lead to potential computational redundancy, resulting in significant computational overhead. TOP does not use this layer-wise inference in experiments, which significantly saves computational costs.

# B   Implementation Details

## B.1   TOP for Variant GNNs

We also extend TOP to the message passing framework for variant message passing-based GNNs. The $l$-th layer of GNNs is defined as

$$\mathbf{h}_i^{(l+1)} = f^{(l+1)} \left( \mathbf{h}_i^{(l)}, \left\{ \left\{ \mathbf{h}_j^{(l)} \right\} \right\}_{j \in \mathcal{N}_i} \right), \tag{9}$$

where $\{\{\dots\}\}$ denotes the multiset. We separate the neighborhood information in Equation (9) of the multiset into two parts

$$\begin{aligned} \mathbf{h}_i^{(l+1)} &= f^{(l+1)} \left( \mathbf{h}_i^{(l)}, \left\{ \left\{ \mathbf{h}_j^{(l)} \right\} \right\}_{j \in \mathcal{N}_i \cap \mathcal{B}} \cup \left\{ \left\{ \mathbf{h}_j^{(l)} \right\} \right\}_{j \in \mathcal{N}_i - \mathcal{B}} \right) \\ &\approx f^{(l+1)} \left( \mathbf{h}_i^{(l)}, \left\{ \left\{ \mathbf{h}_j^{(l)} \right\} \right\}_{j \in \mathcal{N}_i \cap \mathcal{B}} \cup \left\{ \left\{ \mathbf{r}_j \mathbf{H}_{\mathcal{B}} \right\} \right\}_{j \in \mathcal{N}_i - \mathcal{B}} \right), \end{aligned} \tag{10}$$

where $\mathbf{r}_j$ is the $j$-th row of the coefficient matrix $\mathbf{R}$. Equation (10) does not depend on the out-of-batch neighborhood information, achieving a linear computational complexity. We estimate the coefficient matrix $\mathbf{R}$ by

$$\min_{\mathbf{R}} \|\overline{\mathbf{H}}_{\mathcal{N}_{\mathcal{B}}^c} - \mathbf{R}\overline{\mathbf{H}}_{\mathcal{B}}\|_F, \tag{11}$$

We provide more details for the estimation of $\mathbf{R}$ in Appendix B.4.

## B.2   Implementation of GCNII

We follow the implementation[2] of GAS (Fey et al., 2021), which introduces the jumping knowledge connection (Xu et al., 2018) to accelerate the convergence (Xu et al., 2021) for some GNN models.

---

[2]https://github.com/rusty1s/pyg_autoscale

Table 2: **Time and space complexity per gradient update of full-batch gradient descent with whole message passing (Full-batch), CLUSTER (Chiang et al., 2019), GAS (Fey et al., 2021), LMC (Shi et al., 2023), and TOP.**

| Method | Time complexity | GPU Memory | Neighborhood Compensation |
|---|---|---|---|
| Full-batch | $\mathcal{O}(L(\|\mathcal{E}\|d + \|\mathcal{V}\|d^2))$ | $\mathcal{O}(L\|\mathcal{V}\|d)$ | ✓ |
| CLUSTER | $\mathcal{O}(L(deg_{\max}\|\mathcal{B}\|d + \|\mathcal{B}\|d^2))$ | $\mathcal{O}(L\|\mathcal{B}\|d)$ | × |
| GAS and LMC | $\mathcal{O}(L(deg_{\max}\|\mathcal{B}\|d + \|\mathcal{B}\|d^2))$ | $\mathcal{O}(deg_{\max}L\|\mathcal{B}\|d)$ | ✓ |
| TOP | $\mathcal{O}(L(deg_{\max}\|\mathcal{B}\|d + \|\mathcal{B}\|(d^2 + k^2)))$ | $\mathcal{O}(L\|\mathcal{B}\|d)$ | ✓ |

We first run GCNII (Chen et al., 2020) to generate embeddings $\mathbf{H}_{\mathcal{B}}^{(l)}$ for each GNN layer $l$. Then, we compute the final embeddings by the jumping knowledge connection (Xu et al., 2018)

$$\mathbf{H}_{\mathcal{B}}^{final} = MLP^{output}(\frac{1}{L+1}\sum_{l=0}^{L} MLP^{(l)}(\mathbf{H}_{\mathcal{B}}^{(l)})),$$

where MLP is a multi-layer perceptron. We find the best hyperparameters $\alpha, \lambda$ of GCNII by grid search on the Ogbn-products and Ogbn-papers dataset.

### B.3 Hyperparameters

**Comparison with subgraph sampling.** To ensure a fair comparison, we follow the GNN architectures, the data splits, training pipeline, and hyperparameters of GCN and PNA in (Fey et al., 2021). We search the best hyperparameters of GCNII, GAT, and SAGE for TOP, CLUSTER, and GAS in the same set.

**Comparison with node/layer-wise sampling.** We run NS and LABOR by the official implementation[3] of LABOR (Balin & Catalyurek, 2023) and corresponding hyperparameters. For TOP, we uniformly sample nodes to construct subgraphs. To ensure a fair comparison, TOP follows the data splits, training pipeline, learning rate, and hyperparameters of LABOR (Balin & Catalyurek, 2023). We adapt the batch size of TOP such that the memory consumption of TOP is similar to LABOR.

### B.4 Fast Estimation of Coefficient Matrix

We compute the coefficient matrix $\mathbf{R}$ by solving Equation $\mathbf{H}_{\mathcal{N}_{\mathcal{B}}^c}^{(l)} = \mathbf{R}\mathbf{H}_{\mathcal{B}}^{(l)}$. If the size of the subgraph $|\mathcal{B}|$ is large, then solving the linear equation $\mathbf{H}_{\mathcal{N}_{\mathcal{B}}^c}^{(l)} = \mathbf{R}\mathbf{H}_{\mathcal{B}}^{(l)}$ is expensive. As the rank of $\overline{\mathbf{H}}_{\mathcal{B}}$ is less than the hidden dimension $d << |\mathcal{B}|$, there exists a low-rank matrix decomposition such that

$$\overline{\mathbf{H}}_{\mathcal{B}} = \mathbf{Q}\mathbf{Q}^{\top}\overline{\mathbf{H}}_{\mathcal{B}},$$

where $\mathbf{Q} \in \mathbb{R}^{|\mathcal{B}|\times k}$ has orthogonal columns. $k \geq d$ is a hyperparameter. We use $k = d$ in all experiments. We use the proto-algorithm (Halko et al., 2011) to efficiently compute $\mathbf{Q}$. By letting $\hat{\mathbf{R}} = \mathbf{R}\mathbf{Q} \in \mathbb{R}^{|\mathcal{N}_{\mathcal{B}}^c|\times d}$, Equation (11) becomes

$$\min_{\hat{\mathbf{R}}} \|\mathbf{Y}_{\mathcal{B}}(\overline{\mathbf{H}}) - \widetilde{\mathbf{A}}_{\mathcal{B},\mathcal{N}_{\mathcal{B}}^c}\hat{\mathbf{R}}(\mathbf{Q}^{\top}\overline{\mathbf{H}}_{\mathcal{B}})\|_F. \tag{12}$$

Further, we uniformly sample a small set $\mathcal{S}$ with $|\mathcal{S}| = k$ from $\mathcal{B}$ to reduce the costs by

$$\min_{\hat{\mathbf{R}}} \|\mathbf{Y}_{\mathcal{B}}(\overline{\mathbf{H}}) - \widetilde{\mathbf{A}}_{\mathcal{B},\mathcal{N}_{\mathcal{B}}^c}\hat{\mathbf{R}}(\mathbf{Q}_{\mathcal{S}}^{\top}\overline{\mathbf{H}}_{\mathcal{S}})\|_F,$$

which is equivalent to

$$\min_{\hat{\mathbf{R}}} \|\widetilde{\mathbf{A}}_{\mathcal{B},\mathcal{N}_{\mathcal{B}}^c}(\overline{\mathbf{H}}_{\mathcal{N}_{\mathcal{B}}^c} - \hat{\mathbf{R}}(\mathbf{Q}_{\mathcal{S}}^{\top}\overline{\mathbf{H}}_{\mathcal{S}}))\|_F.$$

Since $\overline{\mathbf{H}}_{\mathcal{S}}$ is usually the full-column-rank matrix, we can compute $\hat{\mathbf{R}}$ by $\hat{\mathbf{R}} = \mathbf{H}_{\mathcal{N}_{\mathcal{B}}^c}(\mathbf{Q}_{\mathcal{S}}^{\top}\overline{\mathbf{H}}_{\mathcal{S}})^{\dagger}$ and then save $\partial\hat{\mathbf{A}}_{\mathcal{B}} = \widetilde{\mathbf{A}}_{\mathcal{B},\mathcal{N}_{\mathcal{B}}^c}\hat{\mathbf{R}}$ in the pre-processing phase, where $(\mathbf{Q}_{\mathcal{S}}^{\top}\overline{\mathbf{H}}_{\mathcal{S}})^{\dagger}$ is the Moore-Penrose inverse of

---

[3]https://github.com/dmlc/dgl/tree/master/examples/pytorch/labor

$\mathbf{Q}_\mathcal{S}^\top \overline{\mathbf{H}}_\mathcal{S}$. At the training phase, Equation (6) becomes

$$\mathbf{Z}_\mathcal{B}^{(l+1)} = \widetilde{\mathbf{A}}_{\mathcal{B},\mathcal{B}} \mathbf{H}_\mathcal{B}^{(l)} + \partial \hat{\mathbf{A}}_\mathcal{B} (\mathbf{Q}_\mathcal{S}^\top \mathbf{H}_\mathcal{S}^{(l)}), \tag{13}$$

where $\partial \hat{\mathbf{A}}_\mathcal{B} \in \mathbb{R}^{|\mathcal{B}| \times k}$, $\hat{\mathbf{Q}}_\mathcal{S} \in \mathbb{R}^{k \times k}$, and $\mathbf{H}_\mathcal{S}^{(l)} \in \mathbb{R}^{k \times d}$. The time complexity of the second term in Equation 13 is $\mathcal{O}(|\mathcal{B}|k^2 + k^2 d)$, which is significantly lower than that in Equation (6), i.e., $\mathcal{O}(|\mathcal{B}|^2 d)$, as $|\mathcal{B}| >> d$.

The analysis for message passing-based GNNs is similar.

### B.5 Decrease Approximation Errors by More Basic Embeddings

By concatenating $N$ basic embeddings at different random initialization, we can estimate $\mathbf{R}$ by $(\overline{\mathbf{H}}(\mathcal{W}_1^{(rand)}), \overline{\mathbf{H}}(\mathcal{W}_2^{(rand)}), \ldots, \overline{\mathbf{H}}(\mathcal{W}_N^{(rand)})) \in \mathbb{R}^{n \times (T+1)dN}$, which decreases the approximation errors as $N$ increases.

### B.6 Complexity Analysis

We summarize TOP in Algorithms 1 and 2. TOP first pre-processes the topological compensation by Algorithm 1 and then reuses the topological compensation during the training phase.

---

**Algorithm 1** Pre-processing phase of TOP

---

1: **Input:** Mini-batches $\{\mathcal{B}_i\}_{i=1}^m$
2: Compute $\mathbf{H}^{(l)}$ with a model at random initialization.
3: **for** $i = 1, ..., m$ **do**
4:     Compute $\mathbf{Q}_{\mathcal{S}_i}$ by the proto-algorithm.
5:     Compute $\hat{\mathbf{R}}$ by solving Equation (12).
6:     Compute $\partial \hat{\mathbf{A}}_{\mathcal{B}_i} = \widetilde{\mathbf{A}}_{\mathcal{B}_i, \mathcal{N}_{\mathcal{B}_i}^c} \hat{\mathbf{R}}$
7: **end for**
8: Save $\{\mathbf{Q}_{\mathcal{S}_i}\}_{i=1}^m$ and $\{\partial \hat{\mathbf{A}}_{\mathcal{B}_i}\}_{i=1}^m$
9: **Output:** $\{\mathbf{Q}_{\mathcal{S}_i}\}_{i=1}^m$ and $\{\partial \hat{\mathbf{A}}_{\mathcal{B}_i}\}_{i=1}^m$

---

---

**Algorithm 2** Training phase of TOP

---

1: **Input:** Mini-batches $\{\mathcal{B}_i\}_{i=1}^m$, $\{\mathbf{Q}_{\mathcal{S}_i}\}_{i=1}^m$, and $\{\partial \hat{\mathbf{A}}_{\mathcal{B}_i}\}_{i=1}^m$
2: **for** $i = 1, \ldots, N$ **do**
3:     Randomly sample $\mathcal{B}_i$ from $\{\mathcal{B}_i\}_{i=1}^m$
4:     Initialize $\mathbf{H}_{\mathcal{B}_i}^{(0)} = \mathbf{X}_{\mathcal{B}_i}$
5:     **for** $l = 0, \ldots, L - 1$ **do**
6:         Compute $\mathbf{H}_{\mathcal{B}_i}^{(l+1)} = \sigma(\widetilde{\mathbf{A}}_{\mathcal{B},\mathcal{B}} \mathbf{H}_{\mathcal{B}_i}^{(l)} + (\partial \hat{\mathbf{A}}_{\mathcal{B}_i} (\mathbf{Q}_{\mathcal{S}_i}^T \mathbf{H}_{\mathcal{B}_i}^{(l)})) \mathbf{W}^{(l)})$
7:     **end for**
8:     Compute the mini-batch loss
9:     Update parameters by backward propagation
10: **end for**

---

As the costs of pre-processing are marginal, we compare the computational complexity of the training phase in Table 2. TOP compensates for the neighborhood messages with the least time and memory complexity among existing subgraph sampling methods.

## C More Experiments

### C.1 Measuring Message Invariance in Real-world Datasets.

We conduct extensive experiments on four real-world datasets with five GNN backbones to demonstrate that the message invariance holds in real-world datasets. Table 3 shows that the relative approximation errors of TOP are less than 5% and the test accuracy of TOP is very close to the whole message passing.

Table 3: **Message invariance in real-world datasets.** TOP approximates the whole message passing solely through MP$_{\text{IB}}$ with marginal approximation errors.

| Dataset | GNN | Methods | Relative approximation errors ↓ | | | | | Test accuracy degradation ↓ | | | | |
|---|---|---|---|---|---|---|---|---|---|---|---|---|
| | | | 10% | 20% | 30% | 40% | 50% | 10% | 20% | 30% | 40% | 50% |
| Ogbn-arxiv | GCN | CLUSTER | 23.8% | 20.2% | 18.0% | 15.8% | 12.1% | 5.03% | 3.67% | 2.99% | 2.66% | 1.40% |
| | | GAS | 17.1% | 15.3% | 14.1% | 12.8% | 9.9% | 0.80% | 0.78% | 0.58% | 0.61% | 0.33% |
| | | TOP | **3.5%** | **2.8%** | **2.4%** | **2.2%** | **1.6%** | **0.15%** | **0.10 %** | **0.15%** | **0.12%** | **0.13%** |
| | GCNII | CLUSTER | 13.5% | 11.5% | 10.2% | 9.0% | 6.9% | 4.72% | 3.60% | 2.99% | 2.38% | 1.90% |
| | | GAS | 9.6% | 8.8% | 8.1% | 7.3% | 6.0% | 2.37% | 2.06% | 1.88% | 1.67% | 1.46% |
| | | TOP | **2.5%** | **2.3%** | **2.0%** | **1.8%** | **1.4 %** | **0.42%** | **0.28%** | **0.35%** | **0.17%** | **0.18%** |
| | SAGE | CLUSTER | 15.58% | 13.08% | 11.08% | 9.91% | 7.06% | 4.37% | 3.69% | 3.04% | 2.80% | 1.91% |
| | | GAS | 24.77% | 20.97% | 16.80% | 16.02% | 10.47% | 6.84% | 4.83% | 3.98% | 3.75% | 1.77% |
| | | TOP | **4.38%** | **3.69%** | **3.21%** | **2.89%** | **2.03%** | **0.08%** | **0.11%** | **0.01%** | **0.10 %** | **0.00%** |
| | GAT | CLUSTER | 23.37% | 20.22% | 18.00% | 16.27% | 12.67% | 5.99% | 4.34% | 3.73% | 3.09% | 2.04% |
| | | GAS | 14.96% | 13.47% | 12.51% | 11.53% | 9.08% | 1.17% | 0.97% | 0.89% | 0.80% | 0.67% |
| | | TOP | **3.41%** | **2.99%** | **2.56%** | **2.30 %** | **1.63%** | **0.15%** | **0.16%** | **0.10%** | **0.08%** | **0.08%** |
| Reddit | GCN | CLUSTER | 29.20% | 22.10% | 18.10% | 16.02% | 11.53% | 3.85% | 3.05% | 2.34% | 1.88% | 1.14% |
| | | GAS | 27.13% | 23.87% | 21.07% | 18.69% | 15.00% | 1.27% | 1.22% | 0.82% | 0.68% | 0.49% |
| | | TOP | **0.78%** | **0.65%** | **0.55%** | **0.53%** | **0.39%** | **0.09%** | **0.09%** | **0.08%** | **0.06%** | **0.03%** |
| | GCNII | CLUSTER | 25.32% | 19.62% | 16.45% | 14.94% | 11.29% | 4.79% | 3.59% | 2.45% | 2.78% | 1.46% |
| | | GAS | 32.87% | 30.74% | 27.52% | 26.00% | 23.05% | 7.76% | 5.82% | 4.41% | 4.51% | 2.77% |
| | | TOP | **4.54%** | **4.75%** | **5.22%** | **3.62%** | **2.58%** | **0.33%** | **0.41%** | **0.54%** | **0.31%** | **0.30%** |
| | SAGE | CLUSTER | 10.74% | 7.58% | 6.80% | 5.35% | 3.44% | 3.84% | 2.85% | 2.60% | 2.13% | 1.37% |
| | | GAS | 7.03% | 5.95% | 5.64% | 3.43% | 1.90% | 0.94% | 0.60% | 0.79% | 0.41% | 0.43% |
| | | TOP | **1.31%** | **1.10%** | **1.05%** | **0.85%** | **0.61%** | **0.31%** | **0.27%** | **0.21%** | **0.19%** | **0.11%** |
| | PNA | CLUSTER | 24.13% | 21.40% | 18.60% | 16.70% | 13.60% | 4.16% | 3.20% | 1.76% | 1.43% | 1.13% |
| | | GAS | 22.39% | 20.16% | 18.77% | 16.54% | 13.38% | 2.60% | 2.49% | 1.74% | 1.64% | 0.72% |
| | | TOP | **13.01%** | **11.18%** | **10.48%** | **9.35%** | **7.42%** | **1.48%** | **1.28%** | **0.76%** | **0.97%** | **0.62%** |
| Yelp | GCNII | CLUSTER | 5.74% | 4.48% | 3.82% | 3.25% | 2.38% | 0.89% | 0.57% | 0.45% | 0.34% | 0.21% |
| | | GAS | 7.36% | 6.52% | 5.84% | 5.15% | 4.01% | 1.08% | 0.93% | 0.77% | 0.64% | 0.50% |
| | | TOP | **1.36%** | **1.24%** | **1.14%** | **1.05%** | **0.86%** | **0.13%** | **0.11%** | **0.09%** | **0.08%** | **0.05%** |
| | SAGE | CLUSTER | 15.55% | 12.53% | 10.85% | 9.35% | 6.99% | 1.13% | 0.87% | 0.66% | 0.56% | 0.35% |
| | | GAS | 5.21% | 4.65% | 4.19% | 3.85% | 2.96% | 0.77% | 0.62% | 0.59% | 0.53% | 0.40% |
| | | TOP | **2.41%** | **2.18%** | **2.02%** | **1.84%** | **1.54%** | **0.07%** | **0.07%** | **0.05%** | **0.04%** | **0.04%** |
| Ogbn-products | SAGE | CLUSTER | 9.55% | 8.50% | 7.43% | 6.91% | 5.34% | 1.67% | 1.67% | 1.67% | 1.67% | 1.67% |
| | | GAS | 2.44% | 2.18% | 1.91% | 1.80% | 1.37% | 0.37% | 0.34% | 0.31% | 0.27% | 0.19% |
| | | TOP | **0.86%** | **0.73%** | **0.63%** | **0.58%** | **0.44%** | **0.17%** | **0.14%** | **0.12%** | **0.11%** | **0.11%** |
| **Average** | | CLUSTER | 17.86% | 14.66% | 12.67% | 11.22% | 8.48% | 3.68% | 2.83% | 2.24% | 1.97% | 1.33% |
| | | GAS | 15.54% | 13.88% | 12.40% | 11.19% | 8.83% | 2.36% | 1.88% | 1.52% | 1.41% | 0.88% |
| | | TOP | **3.46%** | **3.05%** | **2.85%** | **2.45%** | **1.86%** | **0.31%** | **0.27%** | **0.22%** | **0.20%** | **0.15%** |

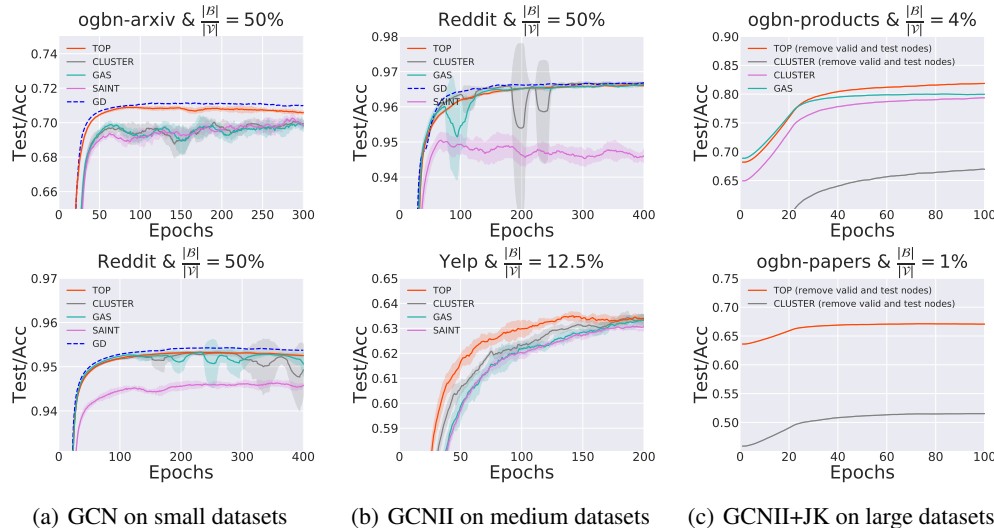

Figure 6: **Convergence curves (test accuracy vs. epoch).** $|\mathcal{B}|$ and $|\mathcal{V}|$ denote the sizes of subgraphs and the whole graph respectively.

## C.2 CONVERGENCE CURVES (TEST ACCURACY VS. EPOCHS)

We provide the convergence curves (test accuracy vs. epochs) in Figure 6. Notably, we report the test accuracy of the full-batch gradient descent (GD) every two steps rather than per epoch, as GD performs backward backpropagation once per epoch while other methods perform backward backpropagation twice per epoch on the Ogbn-arxiv and Reddit datasets. The convergence curves of TOP are close to GD on the Ogbn-arxiv and Reddit datasets, while other subgraph sampling methods fail to resemble the full-batch performance on the Ogbn-arxiv dataset. Moreover, TOP significantly accelerates the convergence on the medium and large datasets, e.g., Yelp, Ogbn-products, and Ogbn-papers.

## C.3 TOP ON ARCHITECTURE VARIANTS

We compare subgraph sampling methods (including TOP, CLUSTER (Chiang et al., 2019), GAS (Fey et al., 2021), SAINT (Zeng et al., 2020), and IBMB (Gasteiger et al., 2022)) on more GNN architectures (i.e., GAT (Veličković et al., 2018) and SAGE (Hamilton et al., 2017)) in Figure 7. TOP is faster than the existing subgraph sampling methods on GAT and SAGE architectures due to its powerful convergence and high efficiency. The experiments demonstrate that TOP is a general framework for different GNN architectures.

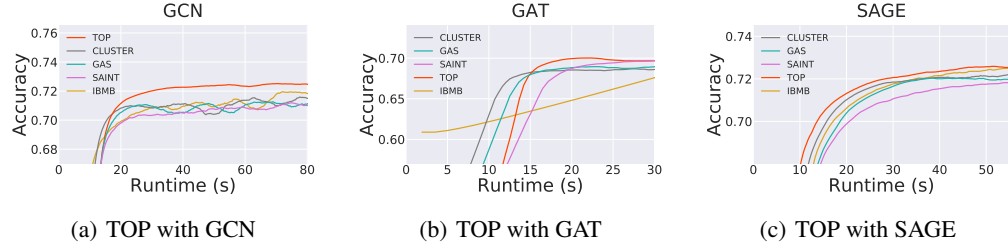

Figure 7: **Convergence curves (test accuracy vs. runtime (s)) on more GNN architectures (i.e., GAT (Veličković et al., 2018) and SAGE (Hamilton et al., 2017)).**

Table 4: **Efficiency of the full-batch gradient descent (Full-batch), GAS, TOP**. $R_*$ and $M_*$ denote the runtime per epoch and memory consumption of the algorithm $*$ respectively.

| Dataset | Model | $\frac{|\mathcal{B}|}{|\mathcal{V}|}$ | $|\mathcal{B}|$ | Runtime (s)↓ | | | $\frac{R_{GAS}}{R_{TOP}}$ | Memory (MB)↓ | | | $\frac{M_{GAS}}{M_{TOP}}$ |
|---|---|---|---|---|---|---|---|---|---|---|---|
| | | | | Full-batch | GAS | TOP | | Full-batch | GAS | TOP | |
| Ogbn-arxiv | GCN | 50.0% | 84672 | 0.63 | 0.35 | **0.33** | 1.07 | 2463.03 | 1566.89 | **1071.27** | 1.46 |
| Reddit | GCN | 50.0% | 116483 | 2.05 | 1.33 | **0.91** | 1.46 | 2796.30 | 2444.25 | **1334.78** | 1.83 |
| Reddit | GCNII | 50.0% | 116483 | 3.21 | 1.94 | **1.49** | 1.30 | 8240.29 | 5509.85 | **3837.94** | 1.44 |
| Yelp | GCNII | 12.5% | 89606 | OOM | 4.29 | **3.98** | 1.08 | OOM | 6752.89 | **2231.76** | 3.03 |
| Ogbn-products | GCNII+JK | 4.0% | 97961 | OOM | 70.68 | **6.04** | 11.70 | OOM | 6574.37 | **1406.60** | 4.67 |

## C.4 RELATIVE RUNTIME PER EPOCH AND RELATIVE MEMORY CONSUMPTION

We report the relative runtime per Epoch and relative Memory Consumption in Table 4. As the graph size increases, the subgraph ratio $|\mathcal{B}|/|\mathcal{V}|$ decreases. TOP enjoys the least runtime and memory consumption among the baselines.

## C.5 COST OF PRE-PROCESSING AND TRAINING

We report the cost of pre-processing and training in experiments in Table 5. On the small and medium datasets (i.e., Ogbn-arxiv, REDDIT, and YELP), the total time of different methods is similar and TOP achieves the least GPU consumption in most experiments. On the large datasets (i.e. Ogbn-products), TOP is significantly faster and more memory-efficient than existing subgraph sampling methods, as it can remove the valid and test nodes from the sampled subgraph without significant performance degradation. Specifically, Equation (6) compensates the neighborhood information from the valid and test nodes based on the mini-batch training nodes. However, the valid and test nodes in subgraphs are important for CLUSTER, as these valid and test nodes are likely to be the neighbors of the training nodes. Directly removing these nodes without any compensation results in significant performance degradation (see Figures 3 and 6). Besides, GAS needs to update the historical embeddings of valid and test nodes many times, leading to expensive computational costs.

Table 5: **The cost of pre-processing and training.**

| GNN & Dataset | Methods | Pre-processing time (s) | Training time (s) | Total Time (s) | Memory (MB) |
|---|---|---|---|---|---|
| GCN & arxiv | GraphSAINT | 0.0 | 122.0 | 122.0 | 1144.1 |
| | CLUSTER | 1.7 | 92.8 | **94.5** | 1312.0 |
| | GAS | 3.0 | 105.0 | 108.0 | 1566.9 |
| | TOP | 5.0 | 99.0 | 104.0 | **1071.3** |
| SAGE & arxiv | GraphSAINT | 0.0 | 114.5 | 114.5 | 1716.2 |
| | CLUSTER | 1.6 | 93.3 | **94.9** | 1450.1 |
| | GAS | 3.0 | 107.8 | 110.7 | 1616.1 |
| | TOP | 5.3 | 98.9 | 104.1 | **1110.2** |
| GAT & arxiv | GraphSAINT | 0.0 | 63.3 | 63.3 | **2060.9** |
| | CLUSTER | 1.7 | 43.7 | **45.4** | 2253.6 |
| | GAS | 3.0 | 52.8 | 55.7 | 3025.9 |
| | TOP | 4.4 | 58.7 | 63.1 | 3177.9 |
| GCN & REDDIT | GraphSAINT | 0.0 | 387.6 | 387.6 | 1398.2 |
| | CLUSTER | 14.9 | 351.7 | **366.7** | 1955.2 |
| | GAS | 16.6 | 532.0 | 548.6 | 2444.3 |
| | TOP | 20.1 | 364.0 | 384.1 | **1334.8** |
| GCNII & REDDIT | GraphSAINT | 0.0 | 672.0 | 672.0 | 3935.2 |
| | CLUSTER | 14.7 | 595.0 | **609.7** | 4242.2 |
| | GAS | 17.4 | 776.0 | 793.4 | 5509.9 |
| | TOP | 21.2 | 596.0 | 617.2 | **3837.9** |
| GCNII & YELP | GraphSAINT | 0.0 | 1648.6 | **1648.6** | 6011.7 |
| | CLUSTER | 12.6 | 1871.6 | 1884.2 | 5940.4 |
| | GAS | 17.3 | 2145.0 | 2162.3 | 6752.9 |
| | TOP | 25.3 | 1990.0 | 2015.3 | **2231.8** |
| GCNII+JK & products | CLUSTER | 35.5 | 3964.4 | 3999.9 | 2048.7 |
| | GAS | 45.5 | 7068.0 | 7113.5 | 6574.4 |
| | TOP | 35.8 | 604.0 | **639.8** | **1406.6** |
| GCNII+JK & papers | CLUSTER | 0.00 | 1007.42 | **1007.42** | **1526.91** |
| | TOP | 144.53 | 1094.01 | 1238.54 | 1560.34 |

Table 6: **Prediction Performance.** Bold font indicates the best result and underline indicates the second best result.

| # nodes
# edges

Method | 169K
1.2M
Ogbn-arxiv
acc↑ | 89K
450K
Flickr
acc↑ | 2.4M
61.9M
Ogbn-products
acc↑ | 111M
1.6B
Ogbn-papers
acc↑ |
|---|---|---|---|---|
| NS-SAGE | 71.49 | 50.10 | 78.70 | 67.06 |
| CLUSTER-GCN | — | 48.10 | 78.97 | — |
| GraphSAINT | — | 51.10 | 79.08 | — |
| SHADOW-GAT | 72.74 | 53.52 | 80.71 | 67.08 |
| SGC | — | 48.20 | — | 63.29 |
| SIGN | — | 51.40 | 80.52 | 66.06 |
| GD-GCNII | **72.83** | 55.28 | OOM | OOM |
| CLUSTER-GCNII | 72.39 | 55.33 | 79.62 | 51.73 |
| GAS-GCNII | 72.50 | **55.42** | 79.99 | OOM |
| TOP-GCNII | $72.52 \pm 0.34$ | $55.21 \pm 0.46$ | $\mathbf{81.96} \pm 0.24$ | $\mathbf{67.21} \pm 0.12$ |

## C.6 PREDICTION PERFORMANCE ON VARIOUS GRAPHS

**Datasets.** We report the prediction performance of TOP on four datasets, i.e., Flickr (Zeng et al., 2020), Ogbn-arxiv, Ogbn-products and Ogbn-papers (Hu et al., 2020), where the two challenging large datasets (i.e., Ogbn-products and Ogbn-papers (Hu et al., 2020)) contain at least 100 thousand nodes and one million edges. As shown by Table 4, as the batch size is significantly lower than the size of the whole graph, the convergence of mini-batch methods under small batch sizes becomes very important.

**Baselines and implementation details.** Our baselines are from the OGB leaderboards (Hu et al., 2020), including node-wise sampling methods (GraphSAGE (Hamilton et al., 2017), subgraph-wise sampling methods (CLUSTER-GCN in the original paper (Chiang et al., 2019), GraphSAINT (Zeng et al., 2020), SHADOW (Zeng et al., 2021) and GAS (Fey et al., 2021)), precomputing methods (SGC (Wu et al., 2019) and SIGN (Rossi et al., 2020)). The GNN backbones of these baselines are different, as more scalable methods usually use more advanced but more memory-consuming GNN backbones. Due to the differences in GNN backbones, frameworks, weight initialization, and optimizers in the baselines, we report CLUSTER-GCNII and GAS-GCNII for ablation studies. The hyperparameter settings are the same as Section 6.1. The results of the baselines are taken from the referred papers and the OGB leaderboards.

**Prediction performance.** We report the prediction performance of TOP in Table 6. On the small datasets (i.e., the ogbn-arxiv and flickr datasets), which have fewer than 170k nodes, the prediction performance of several subgraph sampling methods (CLUSTER, GAS, and TOP) is comparable to gradient descent (GD), as they can use a large subgraph ratio $|\mathcal{B}|/|\mathcal{V}|$ on the small dataset (e.g. 50% used in GAS), such that the sampled subgraphs are close to the whole graph. On the large datasets (i.e., the ogbn-products and ogbn-papers datasets), as the large subgraph ratio may suffer from the out-of-GPU memory issue, we use a small subgraph ratio (less than 4%). However, the small subgraph ratio increases the ratio of missing messages in $\text{MP}_{\text{OB}}$ for CLUSTER and results in the severe staleness issue for GAS (the update of historical embeddings in GAS is infrequent due to very low node sampling probability). The accuracy of TOP is larger than other baselines, as it compensates the messages in $\text{MP}_{\text{OB}}$ by the message-invariant trasformation $g$ and relies solely on up-to-date embeddings, thus avoiding the staleness issue of the historical embeddings.

## C.7 EXPERIMENTS ON HETEROPHILOUS GRAPHS

The message invariance still holds on heterophilous graphs. To verify our claim, we conduct experiments on five heterophilous graphs (i.e., roman-empire, amazon-ratings, minesweeper, tolokers, and questions) provided by the recent heterophilous benchmark (Platonov et al., 2023), as shown in Table 7. We set the subgraph ratio to be 50%, as the heterophilous graphs (10k-50k nodes) are significantly smaller than the homophilic graphs (200k-112000k nodes) in Section 6. On the

heterophilous graphs, although a node may be very different from its neighbors, the neighbors may be similar to other nodes in the subgraph. Notably, the message-invariant transformation in Equation 4 does not restrict that the embedding of an in-batch node should be similar to its neighborhood embeddings, and thus the message-invariant transformation is able to approximate the neighborhood embeddings by other nodes in the subgraph.

Table 7: **Approximation errors of TOP, CLUSTER, and TOP on heterophilous graphs.**

| Heterophilous Graph | CLUSTER | GAS | TOP |
|---|---|---|---|
| amazon-ratings | 4.14% | 1.36% | **1.02%** |
| minesweeper | 54.53% | 19.68% | **3.12%** |
| questions | 20.25% | 9.54% | **4.90%** |
| roman-empire | 5.42% | 1.65% | **0.88%** |

Figure 8 further reports the convergence curves of TOP, CLUSTER, and GAS on the homophily datasets. From Table 7, the approximation errors of TOP are significantly lower than CLUSTER and GAS on minesweeper, tolokers, and questions. Accordingly, TOP is also significantly faster than CLUSTER and GAS on the three datasets.

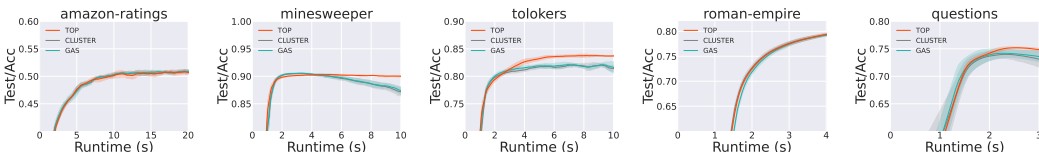

Figure 8: **Convergence curves of TOP, CLUSTER, and GAS on real-world heterophilous graphs.**

## C.8 Experiments under Various Subgraph Samplers

We conduct experiments to demonstrate that TOP consistently brings performance improvement for various subgraph samplers. We first evaluate the relative approximation errors of TOP, CLUSTER, and GAS under METIS, Random, GraphSAINT (Zeng et al., 2020), and SHADOW (Zeng et al., 2021) sampling in Table 8. The results demonstrate that TOP significantly alleviates approximation errors by integrating different subgraph sampling techniques. Specifically, different subgraph sampling techniques are designed to encourage the connections between the sampled nodes with a trade-off for efficiency. METIS aims to directly achieve this goal, while it may be more time-consuming than other sampling techniques. Random sampling is the fastest sampling baseline among them, while it does not consider the connections between the sampled nodes. Thus, Random sampling significantly amplifies the approximation errors of CLUSTER and GAS, while TOP is robust under Random, GraphSAINT, and SHADOW sampling.

Table 8: **Message invariance in real-world datasets with various subgraph samplers.**

| Dataset | GNN | Methods | Relative Approximation Errors ↓ | | | |
|---|---|---|---|---|---|---|
| | | | Random | SAINT | SHADOW | METIS |
| Ogbn-arxiv | GCN | CLUSTER | 30.02% | 15.79% | 12.49% | 12.10% |
| | | GAS | 45.29% | — | — | 9.89% |
| | | TOP | **7.61%** | **5.94%** | **3.41%** | **1.58%** |
| Reddit | SAGE | CLUSTER | 25.91% | 22.22% | 21.32% | 3.44% |
| | | GAS | 13.91% | — | — | 1.90% |
| | | TOP | **3.10%** | **1.45%** | **1.27%** | **0.61%** |
| Yelp | GCNII | CLUSTER | 4.87% | 1.13% | 0.91% | 2.38% |
| | | GAS | 7.86% | — | — | 4.01% |
| | | TOP | **2.68%** | **0.74%** | **0.64%** | **0.86%** |

Figure 9 further shows the convergence curves of TOP, CLUSTER, and GAS under Random sampling. Due to the accurate and fast message passing of TOP, TOP significantly outperforms CLUSTER and GAS in terms of accuracy and converge speeds. By integrating the results with Figures 3(a) and 3(b), the performance improvement of TOP is consistent for various subgraph sampling methods.

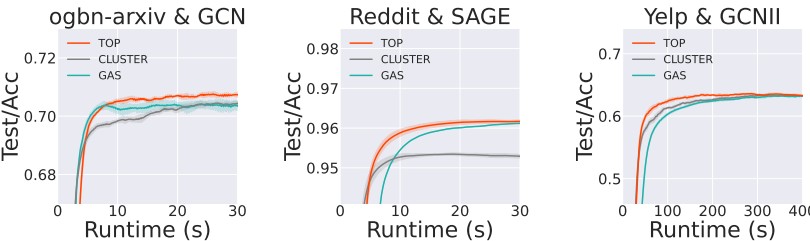

Figure 9: **Convergence curves of TOP, CLUSTER, and GAS under the Random sampler.**

### C.9   DISTANT NODES ARE ESSENTIAL FOR PREDICTION

**Datasets.** We generate a cycle graph with 500,000 nodes and 1,000,000 edges. The node features are randomly sampled from $0, 1$ and the task is to fit the label propagation algorithm, i.e., $\mathbf{X}_t = 0.9 \times \mathbf{A}\mathbf{X}_{t-1} + (1 - 0.9) \times \mathbf{X}_0$ for $t = 1, 2, \ldots, 10$, where $\mathbf{X}_0$ denotes the node features and $\mathbf{A}$ denotes the adjacency matrix. The label $\mathbf{Y}$ is normalized by $\mathbf{X}_{10} / \max_{i,j} [\mathbf{X}_{10}]_{i,j}$. Training, validation, and test node ratios are set at 0.8, 0.1, and 0.1, respectively. We use the mean squared error (MSE) to evaluate the prediction results.

**Model.** We train a ten-layer GCNII to ensure the model can capture sufficient information used in the label propagation algorithm.

**Subgraph sampler.** As the METIS sampling encourages the connections between the sampled nodes, a node, and its most ten-th neighbors appear simultaneously in the subgraph. To remove the high-order information in the subgraph, we instead uniformly randomly sample nodes to construct the subgraph. We partition the whole graph into 200 mini-batches and train three GNNs by CLUSTER, TOP, and Full-batch respectively. CLUSTER trains GNNs using only in-batch nodes and Full-batch trains GNNs using the whole graph.

**Performance.** We run the trained GNNs on the subgraphs induced by $|\mathcal{N}_{\mathcal{B}}^k|$. Table 9 reports the average ratio of subsampled nodes $(1 - |\mathcal{N}_{\mathcal{B}}^k| / |\mathcal{N}_{\mathcal{B}}^{10}|)$ and MSE of the above-mentioned methods.

1. CLUSTER performs significantly worse than Full-batch—which trains GNNs with distant nodes—by order of magnitude ($k = 10$). Thus, the feature vectors of distant nodes are essential for prediction for the ogbn-arxiv dataset.

2. The curve (MSE vs. $k$) of TOP is similar to Full-batch, and both of them achieve the highest MSE at $k = 2$. Thus, TOP's good performance is due to its ability to capture information about distant nodes, similar to Full-batch.

3. The MSE of CLUSTER is lower than TOP and Full-batch when $k \leq 6$, demonstrating that GNNs trained by CLUSTER may overfit to the distribution of sampled subgraphs.

## D   PROOF FOR CONVERGENCE

We first show that TOP based on Equation (5) provides unbiased gradients. We assume that subgraph $\mathcal{B}$ is uniformly sampled from $\mathcal{V}$. When the sampling is not uniform, we use the normalization technique (Zeng et al., 2020) to enforce the assumption.

**Theorem D.1.** *Suppose that the message invariant transformations (4) exist and the subgraph $\mathcal{B}$ is uniformly sampled from $\mathcal{V}$. The iterative message passing of Equations (5) and $\mathbf{H}_{\mathcal{B}}^{(l+1)} = \sigma(\mathbf{Z}_{\mathcal{B}}^{(l+1)}\mathbf{W}^{(l)})$ leads to unbiased mini-batch gradients $\mathbf{d}_{\mathcal{W}}$ such that*

$$\mathbb{E}[\mathbf{d}_{\mathcal{W}}] = \nabla_{\mathcal{W}}\mathcal{L}.$$

Table 9: **Distant Nodes are Essential for Prediction.**

| $k$-hop | $1 - |\mathcal{N}_{\mathcal{B}}^{k}|/|\mathcal{N}_{\mathcal{B}}^{10}|$ | MSE (CLUSTER) | MSE (TOP) | MSE(Full-batch) |
|---|---|---|---|---|
| 0 | 94.99% | 0.021441 | 0.25521 | 0.281997 |
| 2 | 75.22% | 0.004819 | 0.406065 | 0.579234 |
| 4 | 55.84% | 0.000702 | 0.105431 | 0.235721 |
| 6 | 36.85% | 0.001173 | 0.038446 | 0.009939 |
| 8 | 18.24% | 0.001388 | 0.000132 | 0.000049 |
| 10 | 0.00% | 0.001396 | 0.000045 | 0.000040 |

*Proof.* Given any mini-batch $\mathcal{B}$, the embeddings $\mathbf{H}_{\mathcal{B}}^{(l)}$ of TOP are the same as that of SGD due to $\mathbf{H}_{\mathcal{N}_{\mathcal{B}}^{c}}^{(l)} = g(\mathbf{H}_{\mathcal{B}}^{(l)})$ for any GNN parameters $\mathbf{W}^{(l)}$. Thus, their total objective functions of $\mathcal{L} = \mathcal{L}_{TOP}$ are the same.

If TOP is biased, then the expected gradient $\nabla_{\mathcal{W}}\mathcal{L}_{TOP} \neq \nabla_{\mathcal{W}}\mathcal{L}$. Thus, there exists $\epsilon > 0$ and $\mathcal{W}_0$ such that $\mathcal{L}_{TOP}(\mathcal{W}_0) = \mathcal{L}(\mathcal{W}_0)$ while $\mathcal{L}_{TOP}(\mathcal{W}_0 - \epsilon\nabla_{\mathcal{W}_0}\mathcal{L}) \neq \mathcal{L}(\mathcal{W}_0 - \epsilon\nabla_{\mathcal{W}_0}\mathcal{L})$ due to different directional derivatives $\langle\nabla_{\mathcal{W}_0}\mathcal{L}, \nabla_{\mathcal{W}_0}\mathcal{L}_{TOP}\rangle \neq \|\nabla_{\mathcal{W}_0}\mathcal{L}\|_F^2$, which contradicts to $\mathcal{L} = \mathcal{L}_{TOP}$. The unbiasedness holds immediately. $\square$

### D.1 PROOF OF THEOREM 5.1: CONVERGENCE GUARANTEES OF TOP

In this subsection, we give the convergence guarantees of TOP. The proof follows the proof of Theorem 2 in Appendix C.4 of (Chen et al., 2018a).

*Proof.* As $\nabla\mathcal{L}$ is $\gamma$-Lipschitz, we have

$$
\mathcal{L}(\mathcal{W}^{(k+1)})
$$
$$
= \mathcal{L}(\mathcal{W}^{(k)}) + \int_0^1 \langle\nabla\mathcal{L}(\mathcal{W}^{(k)} + t(\mathcal{W}^{(k+1)} - \mathcal{W}^{(k)})), \mathcal{W}^{(k+1)} - \mathcal{W}^{(k)}\rangle \, dt
$$
$$
= \mathcal{L}(\mathcal{W}^{(k)}) + \langle\nabla\mathcal{L}(\mathcal{W}^{(k)}), \mathcal{W}^{(k+1)} - \mathcal{W}^{(k)}\rangle
$$
$$
+ \int_0^1 \langle\nabla\mathcal{L}(\mathcal{W}^{(k)} + t(\mathcal{W}^{(k+1)} - \mathcal{W}^{(k)})) - \nabla\mathcal{L}(\mathcal{W}^{(k)}), \mathcal{W}^{(k+1)} - \mathcal{W}^{(k)}\rangle \, dt
$$
$$
\leq \mathcal{L}(\mathcal{W}^{(k)}) + \langle\nabla\mathcal{L}(\mathcal{W}^{(k)}), \mathcal{W}^{(k+1)} - \mathcal{W}^{(k)}\rangle
$$
$$
+ \int_0^1 \|\nabla\mathcal{L}(\mathcal{W}^{(k)} + t(\mathcal{W}^{(k+1)} - \mathcal{W}^{(k)})) - \nabla\mathcal{L}(\mathcal{W}^{(k)})\|_2 \|\mathcal{W}^{(k+1)} - \mathcal{W}^{(k)}\|_2 \, dt
$$
$$
\leq \mathcal{L}(\mathcal{W}^{(k)}) + \langle\nabla\mathcal{L}(\mathcal{W}^{(k)}), \mathcal{W}^{(k+1)} - \mathcal{W}^{(k)}\rangle + \int_0^1 \gamma t\|\mathcal{W}^{(k+1)} - \mathcal{W}^{(k)}\|_2^2 \, dt
$$
$$
= \mathcal{L}(\mathcal{W}^{(k)}) + \langle\nabla\mathcal{L}(\mathcal{W}^{(k)}), \mathcal{W}^{(k+1)} - \mathcal{W}^{(k)}\rangle + \frac{\gamma}{2}\|\mathcal{W}^{(k+1)} - \mathcal{W}^{(k)}\|_2^2.
$$

Notice that the update formula of $\mathcal{W}^{(k)}$ is

$$
\mathcal{W}^{(k+1)} = \mathcal{W}^{(k)} - \eta\mathbf{d}_{\mathcal{W}}^{(k)},
$$

where $\mathbf{d}_{\mathcal{W}}^{(k)}$ is the gradient of TOP at the $k$-th iteration and we select $\eta < \frac{2}{\gamma}$. Let $\Delta^{(k)} \triangleq \mathbf{d}_{\mathcal{W}}^{(k)} - \nabla\mathcal{L}(\mathcal{W}^{(k)})$, then

$$
\mathcal{L}(\mathcal{W}^{(k+1)})
$$
$$
\leq \mathcal{L}(\mathcal{W}^{(k)}) + \langle\nabla\mathcal{L}(\mathcal{W}^{(k)}), \mathcal{W}^{(k+1)} - \mathcal{W}^{(k)}\rangle + \frac{\gamma}{2}\|\mathcal{W}^{(k+1)} - \mathcal{W}^{(k)}\|_2^2
$$
$$
= \mathcal{L}(\mathcal{W}^{(k)}) - \eta\langle\nabla\mathcal{L}(\mathcal{W}^{(k)}), \mathbf{d}_{\mathcal{W}}^{(k)}\rangle + \frac{\eta^2\gamma}{2}\|\mathbf{d}_{\mathcal{W}}^{(k)}\|_2^2
$$
$$
= \mathcal{L}(\mathcal{W}^{(k)}) - \eta(1 - \eta\gamma)\langle\nabla\mathcal{L}(\mathcal{W}^{(k)}), \Delta^{(k)}\rangle - \eta(1 - \frac{\eta\gamma}{2})\|\nabla\mathcal{L}(\mathcal{W}^{(k)})\|_2^2 + \frac{\eta^2\gamma}{2}\|\Delta^{(k)}\|_2^2.
$$

By taking the expectations of both sides, we have

$\mathbb{E}[\mathcal{L}(\mathcal{W}^{(k+1)})]$

$\leq \mathbb{E}[\mathcal{L}(\mathcal{W}^{(k)})] - \eta(1-\eta\gamma)\mathbb{E}[\langle\nabla\mathcal{L}(\mathcal{W}^{(k)}), \Delta^{(k)}\rangle] - \eta(1-\frac{\eta\gamma}{2})\mathbb{E}[\|\nabla\mathcal{L}(\mathcal{W}^{(k)})\|_2^2] + \frac{\eta^2\gamma}{2}\mathbb{E}[\|\Delta^{(k)}\|_2^2].$

By the properties of the expectations and Theorem D.1, we have

$$\begin{aligned}
\mathbb{E}[\langle\nabla\mathcal{L}(\mathcal{W}^{(k)}), \Delta^{(k)}\rangle] &= \mathbb{E}[\mathbb{E}[\langle\nabla\mathcal{L}(\mathcal{W}^{(k)}), \Delta^{(k)}\rangle|\nabla\mathcal{L}(\mathcal{W}^{(k)})]] \\
&= \mathbb{E}[\langle\nabla\mathcal{L}(\mathcal{W}^{(k)}), \mathbb{E}[\Delta^{(k)}|\nabla\mathcal{L}(\mathcal{W}^{(k)})]\rangle] \\
&= \mathbb{E}[\langle\nabla\mathcal{L}(\mathcal{W}^{(k)}), \mathbb{E}[\mathbf{d}_{\mathcal{W}}^{(k)} - \nabla\mathcal{L}(\mathcal{W}^{(k)})|\nabla\mathcal{L}(\mathcal{W}^{(k)})]\rangle] \\
&= \mathbb{E}[\langle\nabla\mathcal{L}(\mathcal{W}^{(k)}), \mathbb{E}[\mathbf{d}_{\mathcal{W}}^{(k)}|\nabla\mathcal{L}(\mathcal{W}^{(k)})] - \nabla\mathcal{L}(\mathcal{W}^{(k)})\rangle] \\
&= \mathbb{E}[\langle\nabla\mathcal{L}(\mathcal{W}^{(k)}), \nabla\mathcal{L}(\mathcal{W}^{(k)}) - \nabla\mathcal{L}(\mathcal{W}^{(k)})\rangle] \\
&= 0,
\end{aligned}$$

which leads to

$$\mathbb{E}[\mathcal{L}(\mathcal{W}^{(k+1)})] \leq \mathbb{E}[\mathcal{L}(\mathcal{W}^{(k)})] - \eta(1-\frac{\eta\gamma}{2})\mathbb{E}[\|\nabla\mathcal{L}(\mathcal{W}^{(k)})\|_2^2] + \frac{\eta^2\gamma}{2}\mathbb{E}[\|\Delta^{(k)}\|_2^2]$$

$$\Rightarrow \eta(1-\frac{\eta\gamma}{2})\mathbb{E}[\|\nabla\mathcal{L}(\mathcal{W}^{(k)})\|_2^2] \leq \mathbb{E}[\mathcal{L}(\mathcal{W}^{(k)})] - \mathbb{E}[\mathcal{L}(\mathcal{W}^{(k+1)})] + \frac{\eta^2\gamma}{2}\mathbb{E}[\|\Delta^{(k)}\|_2^2].$$

By summing up the above inequalities for $k \in [\![N]\!]$ and dividing both sides by $N\eta(1-\frac{\eta\gamma}{2})$, we have

$$\begin{aligned}
\frac{\sum_{k=1}^N \mathbb{E}[\|\nabla\mathcal{L}(\mathcal{W}^{(k)})\|_2^2]}{N} &\leq \frac{\mathcal{L}(\mathcal{W}^{(1)}) - \mathbb{E}[\mathcal{L}(\mathcal{W}^{(N+1)})]}{N\eta(1-\frac{\eta\gamma}{2})} + \frac{\eta\gamma}{2-\eta\gamma}\frac{\sum_{k=1}^N \mathbb{E}[\|\Delta^{(k)}\|_2^2]}{N} \\
&\leq \frac{\mathcal{L}(\mathcal{W}^{(1)}) - \mathcal{L}^*}{N\eta(1-\frac{\eta\gamma}{2})} + \frac{\eta\gamma}{2-\eta\gamma}\frac{\sum_{k=1}^N \mathbb{E}[\|\Delta^{(k)}\|_2^2]}{N}.
\end{aligned}$$

By noticing that

$$\mathbb{E}[\|\nabla\mathcal{L}(\mathcal{W}^{(R)})\|_2^2] = \mathbb{E}[\mathbb{E}[\|\nabla_{\mathcal{W}}\mathcal{L}(\mathcal{W}^{(R)})\|_2^2 \mid R]] = \frac{\sum_{k=1}^N \mathbb{E}[\|\nabla\mathcal{L}(\mathcal{W}^{(k)})\|_2^2]}{N}$$

and

$$\begin{aligned}
\mathbb{E}[\|\Delta^{(k)}\|_2^2] &= \mathbb{E}[\|\mathbf{d}_{\mathcal{W}}^{(k)} - \nabla\mathcal{L}(\mathcal{W}^{(k)})\|_2^2] \\
&\leq 2(\mathbb{E}[\|\mathbf{d}_{\mathcal{W}}^{(k)}\|_2^2] + \mathbb{E}[\|\nabla\mathcal{L}(\mathcal{W}^{(k)})\|_2^2]) \\
&\leq 4G^2,
\end{aligned}$$

we have

$$\mathbb{E}[\|\nabla\mathcal{L}(\mathcal{W}^{(R)})\|_2^2] \leq \frac{\mathcal{L}(\mathcal{W}^{(1)}) - \mathcal{L}^*}{N\eta(1-\frac{\eta\gamma}{2})} + \frac{4G^2\eta\gamma}{2-\eta\gamma}.$$

If $\eta < \frac{1}{\gamma}$, $\eta = \mathcal{O}(N^{-\frac{1}{2}})$, we have

$$\mathbb{E}[\|\nabla\mathcal{L}(\mathcal{W}^{(R)})\|_2^2] \leq \frac{2(\mathcal{L}(\mathcal{W}^{(1)}) - \mathcal{L}^*)}{N\eta} + 8G^2\eta\gamma = \mathcal{O}(N^{-\frac{1}{2}}).$$

Therefore, by letting $\varepsilon = (\frac{2(\mathcal{L}(\mathcal{W}^{(1)})-\mathcal{L}^*)}{N\eta} + 8G^2\eta\gamma)^{\frac{1}{2}} = \mathcal{O}(N^{-\frac{1}{4}})$, Theorem 5.1 follows immediately. $\square$

# E    ENCODING SYMMETRY IN GRAPHS VIA TOP

In this section, we show that the embeddings of GNNs at random initialization can encode the symmetry in the original graph, which is a specific node similarity.

**Notations** For brevity, $\overline{\mathcal{N}}_i = \mathcal{N}_i \cup \{i\}$ denotes the neighborhood of node $i$ with itself. We recursively define the set of neighborhoods within $k$-hops as $\overline{\mathcal{N}}_i^k = \overline{\mathcal{N}}_{\overline{\mathcal{N}}_i^{k-1}}$ with $\overline{\mathcal{N}}_i^1 = \overline{\mathcal{N}}_i$. For Theorem E.4, we denote all the possible embeddings at the $l$-th layer by $E^{(l)} = \{\mathbf{h}_1^{(l)}, \mathbf{h}_2^{(l)}, \dots, \mathbf{h}_{t^{(l)}}^{(l)}\}$, where $t^{(l)} \leq t$ is the number of different embeddings at the $l$-th layer, $l \in [\![L]\!]$.

**Motivation for the symmetry.** We first motivate the basic embeddings from the graph isomorphism perspective. The 1-dimensional Weisfeiler-Lehman test (i.e., 1-WL test) (Weisfeiler & Leman, 1968) is widely used to distinguish whether two nodes or graphs are isomorphic.

Given initial node feature/representation $h_u^{(0)}$, at the $l$-th iterations, 1-WL test for GCNs updates the node representation $h_i^{(l-1)}$ based on the local neighborhood by

$$h_i^{(l)} = Hash(\{\{h_u^{(l-1)}, u \in \overline{\mathcal{N}}_i\}\}). \tag{14}$$

Following (Xu et al., 2019), we show the connections between GNNs and 1-WL test in Lemma E.1.

Without loss of generality, we present the theories for the GCN version. Extending them to other GNNs is easy.

**Lemma E.1.** *Given a graph $\mathcal{G} = (\mathcal{V}, \mathcal{E})$ and a GNN, if nodes $i, j \in \mathcal{V}$ are indistinguishable under $l$ iterations of the 1-WL test, then there holds*

$$\mathbf{H}_i^{(l)} = \mathbf{H}_j^{(l)},$$

*for all GNN parameters.*

*Proof.* As $i, j$ are indistinguishable under $l$ iterations of 1-WL test, we have $h_i^{(l)} = h_j^{(l)}$. Notice that the function $Hash$ is injective, we have

$$\{\{h_u^{(l-1)}, u \in \overline{\mathcal{N}}_i\}\} = \{\{h_v^{(l-1)}, v \in \overline{\mathcal{N}}_j\}\} \text{ and } |\overline{\mathcal{N}}_i| = |\overline{\mathcal{N}}_j|.$$

Then, we can know that

$$\{\{h_p^{(l-2)}, p \in \overline{\mathcal{N}}_u, u \in \overline{\mathcal{N}}_i\}\} = \{\{h_q^{(l-2)}, q \in \overline{\mathcal{N}}_v, v \in \overline{\mathcal{N}}_j\}\}$$
$$\text{and } \{\{|\overline{\mathcal{N}}_u|, u \in \overline{\mathcal{N}}_i\}\} = \{\{|\overline{\mathcal{N}}_v|, v \in \overline{\mathcal{N}}_j\}\}.$$

which is equivalent to

$$\{\{h_p^{(l-2)}, p \in \overline{\mathcal{N}}_i^2\}\} = \{\{h_q^{(l-2)}, q \in \overline{\mathcal{N}}_j^2\}\} \text{ and } \{\{|\overline{\mathcal{N}}_u|, u \in \overline{\mathcal{N}}_i\}\} = \{\{|\overline{\mathcal{N}}_v|, v \in \overline{\mathcal{N}}_j\}\}.$$

Recursively, we have

$$\{\{h_p^{(l-3)}, p \in \overline{\mathcal{N}}_i^3\}\} = \{\{h_q^{(l-3)}, q \in \overline{\mathcal{N}}_j^3\}\} \text{ and } \{\{|\overline{\mathcal{N}}_u|, u \in \overline{\mathcal{N}}_i^2\}\} = \{\{|\overline{\mathcal{N}}_v|, v \in \overline{\mathcal{N}}_j^2\}\}$$

$$\vdots$$

$$\{\{h_p^{(0)}, p \in \overline{\mathcal{N}}_i^l\}\} = \{\{h_q^{(0)}, q \in \overline{\mathcal{N}}_j^l\}\} \text{ and } \{\{|\overline{\mathcal{N}}_u|, u \in \overline{\mathcal{N}}_i^{l-1}\}\} = \{\{|\overline{\mathcal{N}}_v|, v \in \overline{\mathcal{N}}_j^{l-1}\}\}.$$

By $\mathbf{x}_k = h_k^{(0)}$ and incorporating the equations above, we can know that

$$\{\{(\mathbf{x}_p, \widetilde{A}_{up}), u \in \overline{\mathcal{N}}_i^{l-1}, p \in \overline{\mathcal{N}}_i^l\}\} = \{\{(\mathbf{x}_q, \widetilde{A}_{vq}), v \in \overline{\mathcal{N}}_j^{l-1}, q \in \overline{\mathcal{N}}_j^l\}\}.$$

Thus, we have

$$\mathbf{H}_{\overline{\mathcal{N}}_i^{l-1}}^{(1)} = \sigma(\widetilde{A}_{\overline{\mathcal{N}}_i^{l-1}, \overline{\mathcal{N}}_i^l} \mathbf{X}_{\overline{\mathcal{N}}_i^l} \mathbf{W}^{(0)}) = \sigma(\widetilde{A}_{\overline{\mathcal{N}}_j^{l-1}, \overline{\mathcal{N}}_j^l} \mathbf{X}_{\overline{\mathcal{N}}_j^l} \mathbf{W}^{(0)}) = \mathbf{H}_{\overline{\mathcal{N}}_j^{l-1}}^{(1)}$$

$$\vdots$$

$$\mathbf{H}_{\overline{\mathcal{N}}_i}^{(l-1)} = \sigma(\widetilde{A}_{\overline{\mathcal{N}}_i, \overline{\mathcal{N}}_i^2} \mathbf{H}_{\overline{\mathcal{N}}_i^2}^{(l-2)} \mathbf{W}^{(l-2)}) = \sigma(\widetilde{A}_{\overline{\mathcal{N}}_j, \overline{\mathcal{N}}_j^2} \mathbf{H}_{\overline{\mathcal{N}}_j^2}^{(l-2)} \mathbf{W}^{(l-2)}) = \mathbf{H}_{\overline{\mathcal{N}}_j}^{(l-1)}$$

$$\mathbf{H}_i^{(l)} = \sigma(\widetilde{A}_{i, \overline{\mathcal{N}}_i} \mathbf{H}_{\overline{\mathcal{N}}_i}^{(l-1)} \mathbf{W}^{(l-1)}) = \sigma(\widetilde{A}_{j, \overline{\mathcal{N}}_j} \mathbf{H}_{\overline{\mathcal{N}}_j}^{(l-1)} \mathbf{W}^{(l-1)}) = \mathbf{H}_j^{(l)}$$

for all GNN parameters. $\square$

However, as many GNNs are less expressive than the 1-WL test, it is difficult to find 1-WL isomorphic node pairs by detecting the embeddings in GNNs. Fortunately, TOP does not require as strong expressiveness as the 1-WL test. For two nodes, we do not need to identify whether they are 1-WL isomorphic, but only need to identify whether they are indistinguishable by GNNs.

**Definition E.2.** (Isomorphism under GNNs). Given initial node feature/representation $\mathbf{H}^{(0)}$, at the $l$-th iteration, node pairs $(i, j)$ are *isomorphic* if they are indistinguishable under $l$ iterations of GNNs, i.e. $\mathbf{H}_i^{(l)} = \mathbf{H}_j^{(l)}$ for all GNN parameters.

From the definition E.2, if two nodes are isomorphic under $l$ iterations of GNNs, then their embeddings at the $l$-th layer are the same for all GNN parameters. Therefore, given two indistinguishable nodes under $l$ iterations of GNNs, we can use one to extrapolate the other without any bias.

**Finding isomorphic node pairs.** We estimate the coefficient matrix $\mathbf{R}$ by solving Problem (12) with $\overline{\mathbf{H}}(\mathcal{W}^{rand})$ are the embeddings of GNNs at random initialization. Intuitively, a neural network at random initialization is likely to be a hash function, as it maps different inputs to different vectors in the high dimensional space. The hash function can detect isomorphic node pairs with the same embeddings. We show this by the following theorem.

**Theorem E.3.** *Assume that the activation function $\sigma$ is the LeakyReLU function and GCNs are randomly initialized. If node pairs $(i, j)$ are not isomorphic, then $\mathbf{H}_i^{(l,rand)} \neq \mathbf{H}_j^{(l,rand)}$ with probability one.*

*Proof.* Suppose node pairs $(i, j)$ are not isomorphic. For $l = 1$, we have

$$\sigma(\widetilde{A}_i \mathbf{X} \mathbf{W}^{(0)}) = \mathbf{H}_i^{(1)} \not\equiv \mathbf{H}_j^{(1)} = \sigma(\widetilde{A}_j \mathbf{X} \mathbf{W}^{(0)}).$$

Since the activation function $\sigma = \mathrm{LeakyReLU}$ is injective, we have

$$\widetilde{A}_i \mathbf{X} \mathbf{W}^{(0)} \not\equiv \widetilde{A}_j \mathbf{X} \mathbf{W}^{(0)},$$

leading to

$$\widetilde{A}_i \mathbf{X} \neq \widetilde{A}_j \mathbf{X}.$$

Thus, we have

$$\widetilde{A}_i \mathbf{X} \mathbf{W}^{(0,rand)} \neq \widetilde{A}_j \mathbf{X} \mathbf{W}^{(0,rand)}$$
$$\mathbf{H}_i^{(1,rand)} = \sigma(\widetilde{A}_i \mathbf{X} \mathbf{W}^{(0,rand)}) \neq \sigma(\widetilde{A}_j \mathbf{X} \mathbf{W}^{(0,rand)}) = \mathbf{H}_j^{(1,rand)}$$

for all GCN parameters.

For $l \geq 2$, similar to the case of $l = 1$, we have

$$\widetilde{A}_i \, \sigma(A\mathbf{H}^{(l-2)} \mathbf{W}^{(l-2)}) = \widetilde{A}_i \mathbf{H}^{(l-1)} \not\equiv \widetilde{A}_j \mathbf{H}^{(l-1)} = \widetilde{A}_j \, \sigma(A\mathbf{H}^{(l-2)} \mathbf{W}^{(l-2)}).$$

We only need to prove that $\{\mathbf{W}^{(l-2)} \in \mathbb{R}^{d \times d} \mid \widetilde{A}_i \, \sigma(A\mathbf{H}^{(l-2)} \mathbf{W}^{(l-2)}) = \widetilde{A}_j \, \sigma(A\mathbf{H}^{(l-2)} \mathbf{W}^{(l-2)})\}$ is a Null set in $\mathbb{R}^{d \times d}$. For simplicity, we denote $\alpha^\top = \widetilde{A}_i$, $\beta^\top = \widetilde{A}_j$ and $B = A\mathbf{H}^{(l-2)}$. Thus, $\widetilde{A}_i \, \sigma(A\mathbf{H}^{(l-2)} \mathbf{W}^{(l-2)}) = \widetilde{A}_j \, \sigma(A\mathbf{H}^{(l-2)} \mathbf{W}^{(l-2)})$ is equivalent to $(\alpha - \beta)^\top \sigma(B\mathbf{W}^{(l-2)}) = \mathbf{0}^\top$.

Notice that

$$\sigma(x) = \mathrm{LeakyReLU}(x) = \begin{cases} x, & \text{if } x \geq 0 \\ kx, & \text{if } x < 0 \end{cases},$$

where $k \in \mathbb{R}$ is the negative slope with the default value 1e-2.

Then we can know that

$$(\sigma(B\mathbf{W}^{(l-2)}))_{uv} = \sigma(\sum_{s=1}^{d} B_{us} \mathbf{W}_{sv}^{(l-2)}) = \sum_{s=1}^{d} \sigma_{uv} B_{us} \mathbf{W}_{sv}^{(l-2)},$$

where $u \in [\![n]\!]$, $v \in [\![d]\!]$, $\sigma_{uv} = 1$ or $k$.

Therefore,

$$((\alpha - \beta)^\top \sigma(B\mathbf{W}^{(l-2)}))_v = \sum_{u=1}^n (\alpha_u - \beta_u) \sum_{s=1}^d \sigma_{uv} B_{us} \mathbf{W}_{sv}^{(l-2)}$$

$$= \sum_{s=1}^d \mathbf{W}_{sv}^{(l-2)} \sum_{u=1}^n \sigma_{uv} B_{us} (\alpha_u - \beta_u).$$

Let $\gamma_{sv} = \sum_{u=1}^n \sigma_{uv} B_{us} (\alpha_u - \beta_u) \in \mathbb{R}$ and $\gamma = (\gamma_{sv}) \in \mathbb{R}^{d \times d}$. Then

$$((\alpha - \beta)^\top \sigma(B\mathbf{W}^{(l-2)})) = \mathbf{0}^\top$$

is equivalent to

$$\sum_{s=1}^d \mathbf{W}_{sv}^{(l-2)} \gamma_{sv} = ((\alpha - \beta)^\top \sigma(B\mathbf{W}^{(l-2)}))_v = 0$$

for all $v \in [\![d]\!]$.

However, $\gamma \neq \mathbf{0}$ for all value of $\sigma_{uv}$ since the isomorphism of node pairs $(i, j)$. As a result, for $\sigma_{uv}$ fixed, the solution to $\sum_{s=1}^d \mathbf{W}_{sv}^{(l-2)} \gamma_{sv} = 0$, $v \in [\![d]\!]$ forms a subspace in $\mathbb{R}^{d \times d}$ with the dimension $d \times d - 1$ at most.

Thus, the set $\{\mathbf{W}^{(l-2)} \in \mathbb{R}^{d \times d} \mid \widetilde{A}_i \, \sigma(A\mathbf{H}^{(l-2)}\mathbf{W}^{(l-2)}) = \widetilde{A}_j \, \sigma(A\mathbf{H}^{(l-2)}\mathbf{W}^{(l-2)})\}$ is contained by the union of several subspaces in $\mathbb{R}^{d \times d}$ with the dimension $d \times d - 1$ at most, which means it is a Null set in $\mathbb{R}^{d \times d}$.

Therefore, $\mathbf{H}_i^{(l,rand)} \neq \mathbf{H}_j^{(l,rand)}$ with probability one. $\qquad\qquad\qquad\qquad\qquad\qquad\qquad\square$

From Theorem E.3, for an out-of-batch node $j \in \mathcal{N}_\mathcal{B}^c$, if randomly initialized GCNs detect $i \in \mathcal{B}$ such that $\mathbf{H}_i^{(l,rand)} = \mathbf{H}_j^{(l,rand)}$, then node pairs $(i, j)$ is probably isomorphic. Thus, we can estimate the solution $\mathbf{R}$ to Problem (12) that $\mathbf{R}_j = \mathbf{e}_i^\top$. Moreover, the estimation probably leads to a zero approximation error at node $j$ since $\mathbf{H}_j^{(l)} = \mathbf{e}_i^\top \mathbf{H}_\mathcal{B}^{(l)} = \mathbf{H}_i^{(l)}$ holds for all GCN parameters.

In practice, the ratio of the indistinguishable node pairs increases as the batch size $|\mathcal{B}|$ increases. The following theorem shows that the approximation error of TOP decreases to zero if the batch size $|\mathcal{B}|$ is large enough.

**Theorem E.4.** *Assume that $\mathcal{B}$ is uniformly selected from $\mathcal{V}$, the initial features $\mathbf{X}$ are sampled from a finite set, the number of different embeddings is bounded by $t$, and $|\mathcal{B}| \geq B_0 \triangleq t \log(Lt\varepsilon^{-1})$. Then, there exists the coefficient matrix $\mathbf{R}$ such that $\overline{\mathbf{H}}_{\mathcal{N}_\mathcal{B}^c} = \mathbf{R}\overline{\mathbf{H}}_\mathcal{B}$ with probability $1 - \mathcal{O}(\varepsilon)$ for any GCN parameters.*

*Proof.* By Theorem E.3, if for any out-of-batch node $j$, there exists an isomorphic in-batch node $i \in \mathcal{B}$, then we can easily find the coefficient matrix $\mathbf{R}$ with $\mathbf{R}_i = e_j$ such that $\overline{\mathbf{H}}_{\mathcal{N}_\mathcal{B}^c} = \mathbf{R}\overline{\mathbf{H}}_\mathcal{B}$.

As a result, we only need to estimate the probability of the existence of such an in-batch node $i$. Notice that, if for all $l \in [\![L]\!]$, $\{\{\mathbf{H}_v^{(l)}, v \in \mathcal{B}\}\}$ contains all the embeddings in $E^{(l)}$, then the existence follows immediately. Thus, we estimate the probability of $E^{(l)} \subset \{\{\mathbf{H}_v^{(l)}, v \in \mathcal{B}\}\}, \forall l \in [\![L]\!]$ as a lower bound.

By considering the contrary, for $|\mathcal{B}|$ fixed, we have

$$p(E^{(l)} \subset \{\{\mathbf{H}_v^{(l)}, v \in \mathcal{B}\}\} \mid |\mathcal{B}|) = 1 - p(E^{(l)} \not\subset \{\{\mathbf{H}_v^{(l)}, v \in \mathcal{B}\}\} \mid |\mathcal{B}|)$$

$$= 1 - p(\exists \, \mathbf{h}_u^{(l)} \notin \{\{\mathbf{H}_v^{(l)}, v \in \mathcal{B}\}\} \mid |\mathcal{B}|)$$

$$\geq 1 - \sum_{u=1}^{t^{(l)}} p(\mathbf{h}_u^{(l)} \notin \{\{\mathbf{H}_v^{(l)}, v \in \mathcal{B}\}\} \mid |\mathcal{B}|).$$

Notice that $t^{(l)} \leq t$ and $p(\mathbf{h}_u^{(l)} \notin \{\{\mathbf{H}_v^{(l)}, \; v \in \mathcal{B}\}\} \mid |\mathcal{B}|) = (1 - \frac{1}{t^{(l)}})^{|\mathcal{B}|} \leq (1 - \frac{1}{t})^{|\mathcal{B}|}$, we have

$$
p(E^{(l)} \subset \{\{\mathbf{H}_v^{(l)}, \; v \in \mathcal{B}\}\} \mid |\mathcal{B}|) \geq 1 - \sum_{u=1}^{t^{(l)}} p(\mathbf{h}_u^{(l)} \notin \{\{\mathbf{H}_v^{(l)}, \; v \in \mathcal{B}\}\} \mid |\mathcal{B}|)
$$

$$
\geq 1 - \sum_{u=1}^{t^{(l)}} (1 - \frac{1}{t})^{|\mathcal{B}|}
$$

$$
\geq 1 - t(1 - \frac{1}{t})^{|\mathcal{B}|},
$$

which leads to

$$
p(E^{(l)} \subset \{\{\mathbf{H}_v^{(l)}, \; v \in \mathcal{B}\}\}, \; \forall l \in [\![L]\!] \mid |\mathcal{B}|) = 1 - p(\exists E^{(l)} \not\subset \{\{\mathbf{H}_v^{(l)}, \; v \in \mathcal{B}\}\} \mid |\mathcal{B}|)
$$

$$
\geq 1 - \sum_{l=1}^{L} p(E^{(l)} \not\subset \{\{\mathbf{H}_v^{(l)}, \; v \in \mathcal{B}\}\})
$$

$$
\geq 1 - Lt(1 - \frac{1}{t})^{|\mathcal{B}|}.
$$

By the condition of the batch size $|\mathcal{B}|$, we know that

$$
|\mathcal{B}| \geq t \log(Lt\varepsilon^{-1})
$$

$$
= \frac{\log(Lt\varepsilon^{-1})}{\frac{1}{t}}
$$

$$
\geq \frac{\log(Lt\varepsilon^{-1})}{-\log(1 - \frac{1}{t})}
$$

$$
= -\log_{1-\frac{1}{t}}(Lt\varepsilon^{-1}),
$$

leading to

$$
p(E^{(l)} \subset \{\{\mathbf{H}_v^{(l)}, \; v \in \mathcal{B}\}\}, \; \forall l \in [\![L]\!] \mid |\mathcal{B}|) \geq 1 - Lt(1 - \frac{1}{t})^{|\mathcal{B}|}
$$

$$
\geq 1 - Lt(1 - \frac{1}{t})^{-\log_{1-\frac{1}{t}}(Lt\varepsilon^{-1})}
$$

$$
= 1 - \varepsilon.
$$

Thus, we have

$$
p(E^{(l)} \subset \{\{\mathbf{H}_v^{(l)}, \; v \in \mathcal{B}\}\}, \; \forall l \in [\![L]\!]) = \sum_{|\mathcal{B}|=B_0}^{|\mathcal{V}|} p(E^{(l)} \subset \{\{\mathbf{H}_v^{(l)}, \; v \in \mathcal{B}\}\}, \; \forall l \in [\![L]\!] \mid |\mathcal{B}|) p(|\mathcal{B}|)
$$

$$
\geq \sum_{|\mathcal{B}|=B_0}^{|\mathcal{V}|} (1 - \varepsilon) p(|\mathcal{B}|)
$$

$$
= 1 - \varepsilon.
$$

Therefore, the probability of the existence of the coefficient matrix $\mathbf{R}$ is

$$
p \geq p(E^{(l)} \subset \{\{\mathbf{H}_v^{(l)}, \; v \in \mathcal{B}\}\}, \; \forall l \in [\![L]\!]) \geq 1 - \varepsilon,
$$

which means $p = 1 - \mathcal{O}(\varepsilon)$. $\qquad\square$

*Remark* E.5. The assumption of discrete features in Theorem E.4 is widely used to analyze expressiveness (Xu et al., 2019; Murphy et al., 2019). In real-world graphs with continuous features, finding the exactly indistinguishable node pairs is difficult. For example, the probability of sampling two points with the same value from the Gaussian distribution is zero. Fortunately, Equation (12) still achieves small approximation errors in practice. To verify this claim, we empirically demonstrate that TOP compensates for neighborhood messages well in Figure 2 in Section 5.2.

## F  LIMITATIONS AND BROADER IMPACTS

In this paper, we propose a novel subgraph-wise sampling method to accelerate the training of GNNs on large-scale graphs, i.e., TOP. The acceleration of TOP is due to the assumption of message invariance. We have conducted extensive experiments to demonstrate that the message invariance holds in various datasets. However, it is still possible that the message invariance assumption does not hold in certain datasets and complex GNN models.

Moreover, this work is promising in many practical and important scenarios such as search engines, recommendation systems, biological networks, and molecular property prediction. Nonetheless, this work may have some potential risks. For example, using this work in search engine and recommendation systems to over-mine the behavior of users may cause undesirable privacy disclosure.

