# OpenReview forum: "Accurate and Scalable Graph Neural Networks via Message Invariance"
_ICLR.cc/2025/Conference — ICLR 2025 Poster_

### Official Review · Reviewer_sVt9 · 2024-10-21

**Soundness:** 3
**Presentation:** 3
**Contribution:** 4
**Rating:** 8
**Confidence:** 4

**Summary:**

This paper presents a new strategy for message passing graph neural network applied to sub-graphs. The author defines a new way to approximate the contribution from out of bach neighbors from in batch neighbors. They demonstrate both the accuracy of their method and its speed of convergence through numerous experiments.

**Strengths:**

- The paper is both well-written and well-organized. It finds a good balance between the theoretical settings, simple examples, detailed experiments and adding the necessary details in the appendix.

- The idea of message invariance is simple while yielding impressive results.

- The experiments regarding message invariance are comprehensive, with enough comparisons and demonstrate very impressive results.

- More importantly, the convergence curves offer good results, in particular when knowing about the differences in loss of accuracy with the other methods.

**Weaknesses:**

- Some figures and explanations lack a lot of clarity

- The comparisons with other methods lack more recent algorithms. It seems it would benefit a lot from 1 or 2 extra comparison with a more recent approach.

**Questions:**

- l18: "stores most nodes on the GPU" : in my opinion issues comes more from gradient descent on very large batch of edges rather than nodes?
- l132: "accelerates convergence" with what speed? what complexity?
- Why this choice of norm? Why not using the $p$-Shatten norm which is often used for demonstration related to optimization and matrixes norms.
- l281: Did you made experiments by adding non-linearity in the formulation of $\mathbf{R}$ ?
- l308: How do you decide when to update the embeddings exactly? It would also be nice to add the improvements of accuracy when using updated embeddings in the experiments.
- Can we study what $\mathbf{R}$ looks like?
- l792: Did you try any method related to points cloud such as FPS/FPS++ ?
- Can't we remove $-\mathcal{L}^{*}$ from the inequalities l1323 and below?

**Changes**

- the norm is never defined properly, or no mention of Frobenius is found
- l26: "without accuracy degradation" : seems like it should be "with limited accuracy degradation"
- I find the Figure 1 not very clear, especially without the explanations that come after. I think displaying more clearly the in and out of batch nodes would help a lot.
- Range of integer are usually denotted as $\llbracket 1, L\rrbracket$
- l187: It would be a nice addition to add a footnote that makes $\mathbf{A}^k$ appears in the equality, to explain directly the neighbors explosion
- l224: "expensive costs of [out of batch] neighborhood embeddings"
- l264: Either spend some time to explain why the 2 equality can be considered an equality wrt. $\mathbf{R}$, or change it to an approximation.
- l292/3: Same, or replace "same output"
- Theorem 5.1 l364 is not very well presented, especially regarding the assumptions. I know its surely about the page limits, but the theorem would benefit a lot from 4/5 extra lines for clarity.
- The conclusion doesn't mention any limitations.
- typo in "hyperparameter" l902
- l944: the algorithm doesn't mention how and when recomputing embeddings and $\mathbf{R}$
- l1127: typo in the bold choice for the last 2 raws.
- l1298: The introduction of $\eta$ should also specify we choose  $\eta \neq 2/\gamma$
- l1318: Either add an extra line or a sentence about the re-ordering and telescoping

**Additional comments**

- I find the notation MP-IB, MP-OB a bit difficult to read in the manuscript. Wouldn't something like $MP_{\text{in}}$ be easier?
- I find the notion of "basic" embeddings not very well chosen. Just talk about sampled embeddings, or precise random embeddings if they were used before any training?
- l234 - 244: it feels like we already had enough explanations regarding this and that it could be skipped or skimmed.
- I am unsure about the organization of section 5. It feels like we want to read about the embeddings as fast as possible.
- l286: $\Delta$ is usually used for differences. It feels like the compensation could be mor easily understood as a boundary, and thus denote it $\partial\mathbf{A}$.
- Section B.4 - l900: I think the section would benefit a lot from a few lines to remind what the proto-algorithm is.
- Proof l1268-1274 is not very clear. It could really gain from more explanations.
- l1476-1478 is not very clear. It could really gain from more explanations.

---

### Official Review · Reviewer_oD2v · 2024-10-25

**Soundness:** 2
**Presentation:** 3
**Contribution:** 3
**Rating:** 8
**Confidence:** 4

**Summary:**

This work considers the problem of message-passing in large graphs, where a sampled mini-batch of nodes, $\mathcal{B}$, aggregates information from exponentially growing neighborhood sizes, $\mathcal{N}\_{\mathcal{B}}^c = \mathcal{N}\_{\mathcal{B}} \setminus \mathcal{B}$. To tackle the inefficiency and/or limited performance of existing sampling based methods designed to tackle this problem, a technique called *Tological Compensation* (TOP) is presented, which captures information from out-of-batch neighbors, while limiting the message-passing to the in-batch nodes.

The authors start by hypothesizing an invariant structure in the node representations, such that the hidden representations satisfy $\mathbf{H}^{(l)}\_{\mathcal{N}\_{\mathcal{B}}^c} \approx \mathbf{R} \mathbf{H}^{(l)}\_{\mathcal{B}}$, for some linear transformation $\mathbf{R} \in \mathbb{R}^{|\mathcal{N}\_{\mathcal{B}}^c| \times |\mathcal{B}|}$ learnt during pre-processing. This allows information from $\mathcal{N}\_{\mathcal{B}}^c$ to be efficiently incorporated into a message passing scheme over just $\mathcal{B}$. Specifically, they augment the original propagation matrix as $\tilde{\mathbf{A}}\_{\mathcal{B},\mathcal{B}} + \tilde{\mathbf{A}}\_{\mathcal{B},\mathcal{N}\_{\mathcal{B}}^c} \mathbf{R}$, and then simply propagate messages over $\mathcal{B}$.

On a range of large real-world datasets (169K-111M nodes, 1.1M-1.6B edges), they demonstrate that TOP can stay competitive with the full-batch gradient descent, and outperforms other baselines. They also show that, at convergence, the approximation error in node representations using TOP is less than 5%, adding support for the invariant structure in the node representations. Theoretically, they show that the convergence rate of TOP is $\mathcal{O}(\epsilon^{-4})$, same as node-wise sampling algorithms, and better than $\mathcal{O}(\epsilon^{-6})$ rate for subgraph sampling algorithms.

**Strengths:**

1. The idea of using an invariant structure to approximate messages from out-of-batch nodes to in-batch nodes is unique and interesting.

2. It boasts an $\mathcal{O}(N^{-4})$ convergence rate in the number of iterations $N$, equal to node sampling methods (Section 5.3).

3. The message invariance structure is demonstrated in some dummy cases (Section 4.2), and then validated via experiments (Section 5.2). As simple as a linear approximation is competitive on several large datasets, outperforming several baselines (Section 6).

4. The memory cost of the method scales very slowly with the number of layers in the GNN, unlike other baselines compared (Figure 5).

5. The variance of this method is also low (Figure 3), making it a reliable choice.

**Weaknesses:**

My rating for the work is low right now primarily because the presentation can be significantly improved. I also includes suggestions here, since I think some of them are weaknesses in the manuscript.

1. The citation (Ma & Tang, 2021), near line 46, is a book on graph deep learning, which is too general. I assume you wish to direct the reader to chapter 7 in it. It would be helpful to specify that along with the citation.

2. There seems to be some inconsistency in the definition of the edge set and corresponding matrices: if $(i,j) \in \mathcal{E}$ denotes an edge from $i$ to $j$, then $\mathcal{N}\_i$ is the out-neighborhood of node $i$, i.e. the nodes it *sends messages to* (not receives from), which I believe is not the intended definition. Similarly, $\mathbf{A}\_{ij} = 1$ if $(i,j) \in \mathcal{E}$, then the $i^{\text{th}}$ row of $\mathbf{A}$ whether nodes *receive messages from* node $i$ (which makes the GCN definition in equation (1) wrong). Also, you should mention that $\mathbf{D}$ is the in-degree matrix. Of course, all this can be avoided by assuming you're working with undirected graphs, but it will be more helpful to make the notation consistent for directed graphs.

3. The estimation of topological compensation is arbitrary: Definition 4.1 requires $g$ to satisfy $\mathbf{H}^{(l)}_{\mathcal{N}\_{\mathcal{B}}^c} = g(\mathbf{H}^{(l)}\_{\mathcal{B}})$, $\forall \mathcal{W}$, but the estimation of $\mathbf{R}$ is based on a randomly initialized parameter set. Again, there is no validation for the claim that this method is "very accurate", since the evaluation is on a single set of (learned) parameters. I do see that an alternative protocol proposed is to update $\mathbf{R}$ when the approximation errors increase, but that disagrees with Definition 4.1 (as I understand it, see questions).

4. The assumption of topological invariance in the node representations seems to be very strong, especially for it to hold for all layers ($\forall l\in[L]$) and for all parameter sets ($\forall \mathcal{W}$). It is also not verified in real-world settings. Figure 2(a)-(c) show that the *final node representations* are close to the full-batch representations *at the end of training*, i.e. for the set of learned parameters, and for a single layer; the assumption behind the theoretical motivation is much stronger.

5. About Figure 2, low relative error at the end of training is not sufficient empirical support for the claim of existence of invariance in node features; although, competitive performance on a range of benchmarks does provide a convincing argument. I recommend a rewording of the manuscript such that soundness is prioritized.

6. It is very hard to figure out the exact algorithm behind TOP, eg. how $\mathbf{R}$ is computed, partly because of the typos (see below for a non-exhaustive list). It becomes clear only in Section 5.1 (even then some questions remain). An algorithm block at some point might help with navigation, but I believe some restructuring is needed to improve clarity.

7. The architectures presented in the main text -- GCN and GCNII -- do not represent a wide class of GNNs. It will be more convincing to move the results for one of GAT or GraphSAGE to the main text, and one of GCN or GCNII to the appendix.

8. There is no discussion on the depth of the networks used in Section 6 -- since this method is designed to tackle the problem of exponentially growing neighborhoods, which is especially problematic when using deep networks for large graphs, it is important to understand how the performance is affected by the network depth.

9. For long-range tasks (which are the primary targets for learning with deep GNNs), it is important to capture long-range information. Since TOP estimates the message-invariant transformation $\mathbf{R}$ using only the input features of the batch-neighborhood, followed by message-passing with the 1-neighborhood of the batch, I believe it fails to capture any information from nodes 2-hops or further away (except, possibly, that learnt by $\mathbf{R}$ during pre-processing).

10. Since TOP is expected to have a fast convergence rate, I was hoping to see the plots of loss/accuracy against the number of iterations. The authors do present plots against physical time in Figure 3 and Figure 5, where TOP is competitive or much faster than other algorithms. However, Figure 4 shows that the per-epoch time of TOP is much smaller, implying that in the same physical time, it can get in more optimization steps. For that reason, I am interested in the plots of performance metrics against number of iterations or number of epochs.

**Questions:**

1. Do you have any comments on the regularization effect of stochastic sampling methods, and why TOP outperforms them on test-sets despite (seemingly) not having any regularization mechanism?

2. In Definition 4.1, am I correct to assume that by "for any GNN parameters $\mathcal{W}$", you mean 'for all $\mathcal{W}$' and not 'for a given $\mathcal{W}$'?

3. Definition 4.1 says $g$ needs to satisfy $\mathbf{H}^{(l)}_{\mathcal{N}\_{\mathcal{B}}^c} = g(\mathbf{H}^{(l)}\_{\mathcal{B}})$. Is that supposed to be $\forall l\in[L]$ and $\forall \mathcal{B}\subset \mathcal{V}$?

4. I don't quite understand the argument in the paragraph starting at line 246. Can you elaborate?

5. In section 4.2.2, how valid is the assumption on full column-rank of the convolved feature matrices?

6. The objective near line 262, $\\\|\mathbf{Z}^{(l)}\_{\mathcal{N}\_{\mathcal{B}}^c}-\mathbf{R}\mathbf{Z}^{(l)}\_{\mathcal{B}}\\\|$, what is $l$ here and what is $\mathcal{B}$?

7. The second equality near line 264, that need not be true unless the minimum of the objective $\min\_{\mathbf{R}} \\\|\mathbf{Z}^{(l)}\_{\mathcal{N}\_{\mathcal{B}}^c}-\mathbf{R}\mathbf{Z}^{(l)}\_{\mathcal{B}}\\\| = 0$. Is it 0 because of the full column-rank?

8. In the computation of $\mathbf{R}$ in line 294, what is $\mathcal{B}$ supposed to be?

9. Am I correct to understand that the proof of Theorem 5.1 only uses the unbiasedness of TOP gradients, as shown in Theorem D.1? Doesn't the part after that closely follow the proof of Theorem 2 in Appendix C.4 of Chen et al. (2018a)? If so, this should be emphasized; continue to keep it in the appendix for completeness, though.

10. Possible typos (correct me if I am wrong; in that case, I may have read the related parts wrongly):
  - In definition 4.1, $g$ should be a map from $\mathbb{R}^{|\mathcal{B}|\times d}$ to $\mathbb{R}^{|\mathcal{N}_{\mathcal{B}}^c|\times d}$.
  - In line 258, supposed to be $\mathbf{Z}^{(l)}\_{\mathcal{B}}$ instead of $\mathbf{X}^{(l)}\_{\mathcal{B}}$.
  - The objective near line 262 should be $\\\|\mathbf{Z}^{(l)}\_{\mathcal{N}\_{\mathcal{B}}^c}-\mathbf{R}\mathbf{Z}^{(l)}\_{\mathcal{B}}\\\|$.
  - In line 294, $\bar{\mathbf{H}}\_{\mathcal{N}\_{\mathcal{B}}^c}$ instead of $\bar{\mathbf{H}}\_{\mathcal{N}\_{\mathcal{B}}}$.
  - In line 1269, $\bar{\mathbf{H}}\_{\mathcal{N}\_{\mathcal{B}}^c}$ instead of $\bar{\mathbf{H}}\_{\mathcal{N}\_{(\mathcal{B})^c}}$.
  - In line 1300 (and others below), $\mathbf{d}\_{\mathcal{W}^{(k)}}$ instead of $\mathbf{d}\_{\mathcal{W}}^{(k)}$.
  - Not completely sure, but the second expression in line 1314 seems to be a typo (I can understand what was intended, though).
  - In line 1320 (and following), $\mathcal{W}^{(1)}$ instead of $\mathcal{W}^1$; similarly for $\mathcal{W}^{N+1}$ and $\mathcal{W}^R$.

---

> ### Comment · Reviewer_oD2v · 2024-11-29
>
> > We present the relation between the approximations of final embeddings and message passing-invariant transformations.
>
> I am sorry, I am confused what the point is of this computation. That the bound on embedding error decreases as the message-invariance becomes more and more linear?
>
> > The low relative error indicates that TOP successfully fulfills our objective of minimizing the discrepancy between MP-IB and the whole message passing.
>
> I was under the impression that this was not the desired objective, but an underlying assumption behind the methodology.
>
> > We present the algorithm block in Algorithms 1, and 2 in Appendix B.5.
>
> Thank you! It would be ideal to add a link to these blocks in the main text (preferably early).
>
> Most of my other concerns were resolved. I have updated my rating. I would appreciate clearer writing in the camera-ready version where the motivations, claims and results are not overstated; highlight whenever you deviate from the theoretical model, and clearly state the approximations made.

---

> > ### Author Response · Authors · 2024-11-29
> > **Thanks for your kind support and for helping us improve the paper!**
> >
> > Dear Reviewer oD2v,
> >
> >
> > Thanks for your kind support and for helping us improve the paper! Below, we provide detailed responses to your further comments.
> >
> >
> >
> > **1. That the bound on embedding error decreases as the message invariance becomes more and more linear?**
> >
> >
> > Yes. In Section 4.2, we provide two examples where the message invariance is linear, and in these cases, the corresponding approximation errors of embeddings are indeed zero.
> >
> >
> >
> > **2. I was under the impression that this was not the desired objective, but an underlying assumption behind the methodology.**
> >
> >
> > Thanks for the question. The underlying assumption of our methodology is that message invariance  in practice is approximately linear. Based on this assumption, the practical implementation of TOP—using linear regression to approximate the message-invariant transformation—significantly reduces the discrepancy between MP-IB and the whole message-passing process in real-world datasets (Figure 2 and Table 3).
> >
> >
> >
> >
> > **3. It would be ideal to add a link to these blocks in the main text (preferably early).**
> >
> >
> > Thanks for your valuable suggestions. We will add the link to Algorithms 1 and 2 in the main text in the camera-ready version if the paper is accepted.
> >
> >
> >
> >
> > Once again, we deeply appreciate your invaluable guidance and the time and effort you have devoted to reviewing our paper.
> >
> > Best,
> >
> > Authors

---

### Official Review · Reviewer_xxsU · 2024-10-26

**Soundness:** 3
**Presentation:** 2
**Contribution:** 3
**Rating:** 3
**Confidence:** 3

**Summary:**

The authors propose a new graph subsampling technique based on the concept of message invariance. They describe message invariance in definition 4.1: which states that the feature vectors of nodes that are not inside the current mini-batch can be recovered via some unknown transformation of the feature vectors in the subsampled batch.

**Strengths:**

- Message invariance seems like an interesting new idea.
- The authors give a couple of theoretical examples for simple problems (using linear GNNs) in which message invariance holds. I understand that generalizing this to include non-linearities can be very difficult.
- Their method is more computationally efficient than other baselines.

**Weaknesses:**

- I am unsure whether message-invariance exists. The method proposed by the authors, see for example equation 6, could also be interpreted as a graph rewiring of the subsampled nodes of the induced subgraph. Good performance could come from rewiring and alleviation of other known issues in GNNs such as over-squashing for instance, rather than due to learning message-invariance.
- The large graph datasets that the authors use tend to be quite local. Rather than message-invariance, it could just be the case that the features of far away nodes are irrelevant for the final prediction (there is no long-range dependency). Hence using only features of nearby nodes may be enough.
- It would be interesting to see how this technique affects the performance of graph transformers such as SGFormer (https://arxiv.org/pdf/2306.10759) which are designed for large scale graphs and which allow for global communication.
- This is minor, but the first time I read the abstract I found it quite confusing, particularly the second sentence. Also, mentioning in the abstract that your method is particularly targeting large graph transductive learning would be good.

**Questions:**

- Why is it reasonable to assume that the transformation g (related to message invariance) can be approximated by a linear projection (as stated in line 280)? If such a transformation exists, wouldn't it likely be highly non-linear? Did this originate based on the linear GNN example? Is there a way of at least learning a non-linear transformation and letting the network figure it out by itself rather than imposing linearity by construction? 1) Provide more justification for why a linear approximation is reasonable, beyond just the linear GNN example 2) Discuss the limitations of assuming linearity and how this may impact performance 3) Consider exploring non-linear transformations and compare performance to the linear case

- Some baselines in the large graph representation learning literature train on stochastically subsampled graphs on GPU, but at inference time they do prediction using all nodes in the graph on CPU. In such cases, the model would actually see out of batch nodes at inference time. Any thoughts? 1) Discuss how your method compares to or could be adapted for approaches that use full graphs at inference time, 2) Consider evaluating your method in both subsampled and full graph inference settings


- Could the authors conduct a synthetic experiment in a controlled setting where they ensure that feature vectors of distant nodes are essential for prediction and empirically demonstrate the following:

   1. Train the model using only in-batch nodes and demonstrate that this approach performs significantly worse than training with distant nodes (which, due to subsampling, are likely out of batch).

   2. Train with the proposed framework and test the number of subsampled nodes and the average n-hop distance required to include in-batch nodes to recover performance.

I would suggest running this on graphs with a manageable number of nodes that can fit on a GPU, yet are sparse and high in diameter.

I would be happy to raise my score if the authors can show message invariance in a more controlled environment.

---

> ### Comment · Reviewer_xxsU · 2024-11-28
>
> ## 1. Good performance could come from rewiring and alleviation of other known issues in GNNs such as over-squashing for instance, rather than due to learning message-invariance.
>
> I agree that altering the connectivity within the subgraph induced by B cannot resolve over-squashing across the entire graph. However, it can mitigate over-squashing within the subgraph used for message passing. If the information remains sufficiently local and the local structure is rewired effectively, this approach can play a crucial role in improving the final prediction, particularly when the graph lacks long-range dependencies.
>
> The global attention mechanism in SGFormer is linearized and does not apply the softmax function. The relative improvement in SGFormer’s performance compared to its parallel message-passing backbone is minimal; in fact, you might observe that disabling the attention has little to no impact on performance. Moreover, your results show a performance difference of less than 1% between the training paradigms (CLUSTER vs. TOP), which does not appear to be statistically significant. Additionally, the datasets used in your experiments do not inherently exhibit long-range dependencies

---

> ### Comment · Reviewer_xxsU · 2024-11-28
>
> ## 2. The synthetic experiment in a controlled setting to ensure that feature vectors of distant nodes are essential for prediction.
>
> I would like to thank the authors for the experiments, but this is not a synthetic dataset. I meant to engineer a signal over a graph which would be handcrafted to be long-range. Ogbn-arxiv is not a long-range dependency dataset.
>
> Furthermore, the model being used in your experiments is designed with 4 message-passing layers. By construction, this limits the model to performing message-passing within local neighborhoods, resulting in a receptive field equivalent to the number of layers—specifically, 4 hops. It is worth noting that a 4-hop neighbor is not considered a "distant" hop.

---

> ### Comment · Reviewer_xxsU · 2024-11-28
> **Response to point 4**
>
> ## 4. Provide more justification for why a linear approximation is reasonable, beyond just the linear GNN example.
>
> I am familiar with Taylor series expansions, but I find no compelling reason to assume that a linear approximation is reasonable, aside from considerations of computational cost. Furthermore, if such message invariance were to exist, I would expect the transformation to be highly complex and decidedly non-linear.

---

> ### Comment · Reviewer_oD2v · 2024-11-29
>
> > I am unsure whether message-invariance exists.
>
> Jumping in to give my view on this problem. I do agree that a linear message-invariance is likely unrealistic, and the experimental results do not verify the existence of it. From what I understand, what is verified (in Figure 2a-c) is that at convergence, the linear approximation reasonably holds. (Perhaps this is an artefact of training with TOP, but I am not sure.) This does not imply that linear message-invariance enables performative training with TOP, but rather that it is an outcome of training with TOP.
>
> Importantly, the invariance assumption would likely have implications beyond this problem, and it is unethical to make such a strong claim without proper verification.
>
> That being said, I believe the experimental results provide sufficient evidence for the efficacy of the algorithm. For that reason, my recommendation of accepting this work comes with a request to the authors to reword the manuscript to clearly indicate that the linear approximation is an approximation/heuristic. It is better to indicate that you have *assumed* such an invariance to exist in practice, based on the examples you present, and have accordingly designed your algorithm.

---

> > ### Author Response · Authors · 2024-12-01
> > **Many thanks for your kind support!**
> >
> > Many thanks for your kind support! We are trying our best to revise our submission following your valuable suggestions and conduct the experiments to address the further concerns raised by Reviewer xxsU. Your guidance is greatly appreciated!

---

### Official Review · Reviewer_74yw · 2024-11-06

**Soundness:** 3
**Presentation:** 3
**Contribution:** 3
**Rating:** 6
**Confidence:** 2

**Summary:**

The paper presents Topological Compensation (TOP), a method for improving mini-batch scalability in Graph Neural Networks (GNNs) by leveraging message invariance. Traditional mini-batch methods often face computational challenges due to dependency on out-of-batch neighbors, especially with deeper GNN layers. The proposed approach replaces costly message-passing operations between in-batch and out-of-batch nodes with a mThe concept of message invariance is innovative and addresses a critical bottleneck in GNN mini-batch processing. Converting MP-OB to MP-IB is well-motivated and theoretically sound. The experimental validation is robust, covering various GNN models (e.g., GCN, GAT) and datasets with different characteristics. essage-invariant transformation that effectively approximates these operations without sacrificing accuracy.

**Strengths:**

The concept of message invariance is innovative and addresses a critical bottleneck in GNN mini-batch processing. Converting MP-OB  to MP-IB is well-motivated and theoretically sound.

**Weaknesses:**

For large datasets, transformers based architechtures are very used for these datasets. Can your theoritical analysis be extended to GNN which doesnt follow the message passing scheme, for example GraphGPS?

**Questions:**

1- Would TOP benefit from combining message invariance with other sampling techniques to further improve scalability or accuracy?

---

### Meta-Review · Area_Chair_qdC9 · 2024-12-23

**Metareview:**

This paper proposes a topological compensation (TOP) method to address the issue of missing higher-order out-of-batch neighbors in mini-batch GNN training, and shows promising experimental results. While uncertainty remains about how accurate the linear approximation to message invariance is in practice, existing results are already good and shows the efficacy of TOP. Therefore, I recommend accept.

**Additional Comments On Reviewer Discussion:**

The main concern comes from reviewer xxsU which questions how accurate the linear approximation is in practice, and looks for results on controlled synthetic datasets. The authors have provided the results later in the rebuttal which should address the concerns.

---

### Decision · Program_Chairs · 2025-01-22

Accept (Poster)